# Hierarchical Frequency Tagging Probe (HFTP): A Unified Approach to Investigate Syntactic Structure Representations in Large Language Models and the Human Brain

**Jingmin An**[1], **Yilong Song**[1], **Ruolin Yang**[1], **Nai Ding**[2], **Lingxi Lu**[3]
**Yuxuan Wang**[1], **Wei Wang**[4], **Chu Zhuang**[1], **Qian Wang**[1,†], **Fang Fang**[1,†]
[1]Peking University, [2]Zhejiang University, [3]Beijing Language and Culture University,
[4]Beijing Institute for General Artificial Intelligence
anjm@stu.pku.edu.cn, {wangqianpsy, ffang}@pku.edu.cn

## Abstract

Large Language Models (LLMs) demonstrate human-level or even superior language abilities, effectively modeling syntactic structures, yet the specific computational units responsible remain unclear. A key question is whether LLM behavioral capabilities stem from mechanisms akin to those in the human brain. To address these questions, we introduce the Hierarchical Frequency Tagging Probe (HFTP), a tool that utilizes frequency-domain analysis to identify neuron-wise components of LLMs (e.g., individual Multilayer Perceptron (MLP) neurons) and cortical regions (via intracranial recordings) encoding syntactic structures. Our results show that models such as GPT-2, Gemma, Gemma 2, Llama 2, Llama 3.1, and GLM-4 process syntax in analogous layers, while the human brain relies on distinct cortical regions for different syntactic levels. Representational similarity analysis reveals a stronger alignment between LLM representations and the left hemisphere of the brain (dominant in language processing). Notably, upgraded models exhibit divergent trends: Gemma 2 shows greater brain similarity than Gemma, while Llama 3.1 shows less alignment with the brain compared to Llama 2. These findings offer new insights into the interpretability of LLM behavioral improvements, raising questions about whether these advancements are driven by human-like or non-human-like mechanisms, and establish HFTP as a valuable tool bridging computational linguistics and cognitive neuroscience. This project is available at `https://github.com/LilTiger/HFTP`.

## 1 Introduction

Language is fundamental to human communication, thought, and cultural transmission. According to the framework proposed by Noam Chomsky, language is divided into three key components: semantics (meaning), phonology (sound), and syntax (hierarchical sentence structure) [Chomsky, 1965]. Syntax is particularly crucial as it governs how words combine into meaningful expressions, underpinning the recursive and generative capacity unique to human language. The theory of Universal Grammar proposes that all human languages share innate structural principles [Chomsky, 1980]. Building on this foundation, cognitive neuroscience has shown that syntactic processing recruits mechanisms distinct from other linguistic functions, particularly within the left inferior frontal and posterior temporal regions [Matchin and Wood, 2020, Friederici, 2011]. Moreover, increases in syntactic complexity yield graded activation in left inferior frontal and posterior temporal cortices, consistent with the computation and maintenance of hierarchical dependencies rather than simple lexical associations [Pallier et al., 2011, Nelson et al., 2017]. As our understanding of human syntactic

computation deepens, artificial intelligence models have increasingly sought to emulate this ability to capture and represent structured language.

In recent years, large language models (LLMs) have evolved rapidly, achieving human-level or better performance on a range of linguistic benchmarks and professional exams [Achiam et al., 2023]. Their success in understanding, translation, and summarization has led to claims of human-like fluency, particularly in generating text that conforms to surface syntactic regularities [Mahowald et al., 2024, He et al., 2024, Van Veen et al., 2024]. Yet it remains unclear whether such models truly represent the hierarchical sentence structures that characterize human syntax. Some findings indicate that LLMs can implicitly capture and manipulate structural relations [Manning et al., 2020], while others suggest their success can rely on shallow statistical heuristics rather than genuine structural understanding [Linzen et al., 2016, McCoy et al., 2019]. This ongoing debate underscores the need for a unified analytic framework capable of directly comparing syntactic representations in human and model systems, which is essential for probing the depth and nature of syntactic alignment between the human brain and artificial models.

Ding et al. [Ding et al., 2016] introduced the hierarchical frequency tagging (HFT) technique to uncover how the human brain processes hierarchical linguistic structures during natural speech comprehension. In this paradigm, monosyllabic words are presented at a rate of 4 Hz to form phrases at 2 Hz, which combine into sentences at 1 Hz. Using frequency-domain analysis of electrophysiological signals, Ding et al. deconstruct the processing of linguistic structures such as phrases and sentences. Subsequent work has extended the HFT framework along complementary axes. Attention is required to group lower-level inputs into higher-order linguistic units, and diverting attention attenuates word- and phrase-rate tracking [Ding et al., 2018]. MEG source analyses dissociate cortical signatures for word- versus phrase-level rhythms and link phrasal tracking to comprehension [Keitel et al., 2018]. Computational modelling shows that oscillatory architectures can implement hierarchical parsing and reproduce HFT-like spectra [Martin and Doumas, 2017]. Natural-speech EEG reveals endogenous word-rate tracking that interacts with exogenous rhythmic cues [Luo and Ding, 2020]. Naturalistic experiments show that phrase-rate tracking reflects internally generated structure rather than compositional meaning per se, yet remains sensitive to the lexical–syntactic cues that support structure building [Coopmans et al., 2022]. Together, these studies demonstrate that HFT effectively isolates neural markers of hierarchical language processing.

Building on hierarchical frequency-tagging [Ding et al., 2016], we introduced the Hierarchical Frequency-Tagging Probe (HFTP), a unified framework that probes internal representational structure and systematically assesses the alignment of syntactic representations between LLMs and the human brain. The key contributions of this paper are: **(i)** We innovatively employed frequency-domain analysis using HFTP to characterize the syntactic structure representations of every computational unit in each layer of LLMs; **(ii)** HFTP provides a simple, universally applicable approach for detecting and aligning syntactic structure representations in LLMs (via neuron-wise probing) and the human brain (via population-level analyses), and extends seamlessly to naturalistic text. **(iii)** Using syntactic templates derived from HFTP, we identified brain regions highly correlated with LLMs, predominantly located in key language-processing areas of the left hemisphere; **(iv)** By comparing six LLMs, we observed divergent trends in upgraded versions, with some showing increased similarity to brain representations while others exhibited reduced alignment. In sum, HFTP effectively detects syntactic structure representations in both LLMs and the human brain, providing a novel framework for alignment study.

## 2 Related work

**Syntactic processing in the human brain** In humans, syntactic processing recruits a left-dominant fronto-temporal network that supports hierarchical combination from finite elements. A series of classic lesion and neuroimaging work documents a left-hemisphere advantage [Friederici and Brauer, 2009, Hagoort, 2013, Blank et al., 2016], with hemispheric temporal sensitivities aiding speech segmentation [Albouy et al., 2020]. Converging evidence shows that syntactic operations are distributed across frontal and temporal cortex with substantial overlap with semantic integration [Blank et al., 2016, Fedorenko et al., 2020]. Artificial-grammar fMRI further indicates that hierarchically structured strings reliably engage left inferior frontal gyrus and posterior superior temporal regions [Chen et al., 2021]. Overall, syntactic processing relies on a left-dominant,

distributed fronto–temporal network that substantially overlaps with semantic integration, rather than a single locus.

**Syntactic processing in language models** Early work shows that LSTM language models are able to capture syntax–sensitive dependencies, such as the phenomenon of subject–verb agreement [Linzen et al., 2016, Kuncoro et al., 2018]. Using the structural probe, a landmark study demonstrates that transformer-based models such as BERT encode hierarchical syntactic trees, enabling these models to represent complex syntax without explicit supervision [Hewitt and Manning, 2019]. Transformer-based models excel at tracking both local and long-range dependencies through specialized attention mechanisms and distribute syntactic knowledge across layers [Clark, 2019, Tenney et al., 2019, Manning et al., 2020]. However, methodological choices in these language-model studies make it challenging to directly relate the findings to human brain activity.

**Alignment between LLMs and the human brain** A growing body of work shows that sentence-level contextual embeddings from predictive language models strongly predict cortical responses during comprehension [Sun et al., 2020, Schrimpf et al., 2021]. Disentanglement analyses that factorize activations into lexical, compositional, syntactic, and semantic components indicate distributed contributions—often with compositional/lexical signals explaining much of the alignment, rather than syntax alone [Caucheteux et al., 2021, Caucheteux and King, 2022]. Neuroimaging manipulations that dissociate semantics from syntax reveal distinct patterns (including frontal engagement) without establishing syntactic dominance [Wang et al., 2020]. Intervention studies that selectively remove linguistic properties from model representations yield reliable drops in brain alignment, with syntactic properties (e.g., tree depth, top constituents) exerting large cross-layer effects [Oota et al., 2023]. These findings are consistent with convergence on shared representational axes across brains and models [Hosseini et al., 2024] and with demonstrations that model-derived stimuli can causally drive or suppress the human language network [Tuckute et al., 2024], while alignment varies systematically across layers as contextual information accrues [Goldstein et al., 2022]. However, methodological heterogeneity—spanning representation choices, tokenization, brain–model alignment metrics, and control analyses—precludes systematic comparison of how syntactic structure is encoded across models and neural populations. A unified analytical framework is therefore needed to harmonize methodological dimensions and support comparable, interpretable cross-system evaluations.

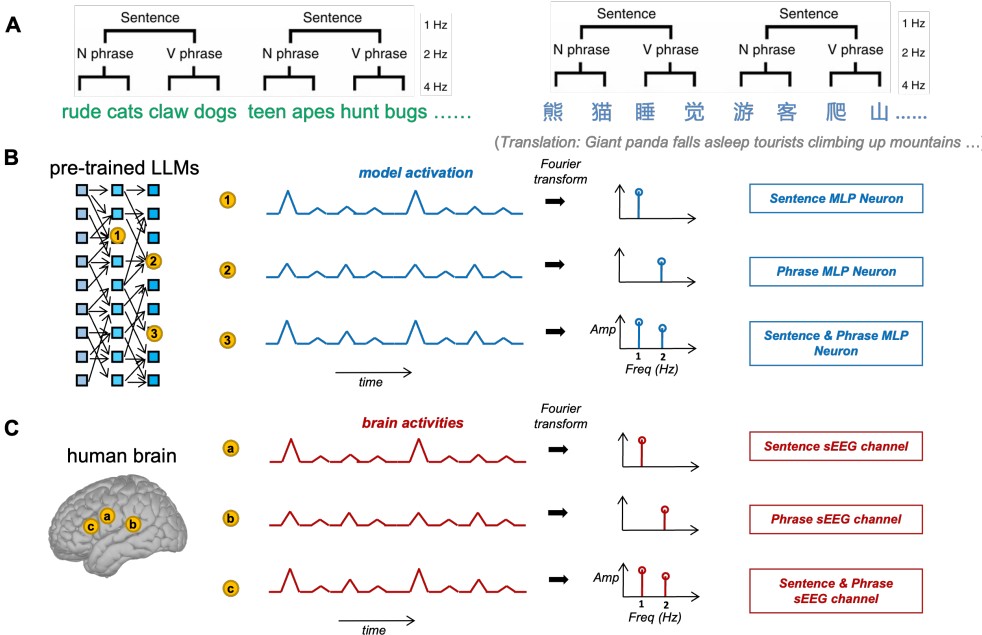

Figure 1: A framework for Hierarchical Frequency Tagging Probes (HFTP) and an illustration of computational units involved in different levels of hierarchical linguistic processing in both LLMs and the human brain. **A**, hierarchical linguistic structure in English and Chinese including syllable, phrase, and sentence. **B**, hierarchical linguistic pattern (1 Hz: sentence feature, 2 Hz: phrase feature) observed both in LLMs and **C**, human brain.

# 3 Methods

We present the framework of the proposed HFTP methodology (see Figure 1). This framework is organized into four parts: Section 3.1 describes the syntactic corpora used and the LLM architectures; Section 3.2 details the syntactic structure probe in LLMs to detect syntactic units; Section 3.3 explains the syntactic structure probe applied to human intracranial stereo-electroencephalography (sEEG) data; and Section 3.4 correlates syntactic structure representations in LLMs and the human brain by comparing frequency-domain representations and detecting similarities in how syntactic structures are encoded across both systems.

## 3.1 Data and LLMs

We mainly utilized Chinese and English corpora adapted from [Ding et al., 2016], consisting of four-syllable sequences in Chinese or four-word sequences in English, where the first two and last two units form phrases (see Figure 1). Further details regarding the corpus can be found in Appendix H. We also adopted naturalistic texts to test the generalizability of our HFTP method (see E). For both the sEEG and model–brain alignment experiments, we used the same two Chinese corpora—the **sentence** and **phrase** corpora—from [Sheng et al., 2019]. While these corpora share a similar structure to the Chinese syntactic corpus used in the LLM experiments, they differ in content. Guided by evidence that periodic lexical regularities alone can produce peaks at word-, phrase-, and sentence-rate frequencies [Frank and Yang, 2018], we added a within-sentence word-order–randomized control in all experiments: items are permuted across positions so that lexical and part-of-speech categories do not recur at fixed positions and any consistent phrase- or sentence-level patterns are prevented, while the lexical set is preserved. This control isolates the syntactic and lexical contribution to 1, 2, and 4 Hz power.

We applied HFTP to six LLMs of varying architectures and sizes—GPT-2, Gemma, Gemma 2, Llama 2, Llama 3.1, and GLM-4 (see Table 5). Note that the GPT-2 model we used is Chinese-pre-trained and supports both Chinese and English. To avoid tokenization artefacts, we average MLP activations over sub-tokens at the syllable level for Chinese and word level for English. This ensures that the 1 Hz and 2 Hz spectral components reflect linguistic boundaries rather than tokenization boundaries, enabling consistent cross-lingual and cross-model comparisons. Notably, the term "MLP neuron" denotes a computational unit in the intermediate hidden layer of the MLP sub-layer within a Transformer model. This sub-layer consists of two linear transformations separated by a nonlinear activation function. We target MLP neurons because they house localized, interpretable units—conceptualized as "knowledge neurons" [Dai et al., 2021] that causally control factual recall and lexical–syntactic concepts [Geva et al., 2022]. This mechanistic specificity provides the discrete, concept-aware handles required by HFTP to robustly localize syntactic structures.

## 3.2 Syntactic structure probe in LLMs

For each LLM, sequences from the Chinese and English corpora were concatenated separately into continuous texts to capture neural-like activations. During this process, each Chinese syllable (or English word) outputs an activation value, allowing the signal corresponding to every individual linguistic unit to be traced. These time-domain activations were then transformed into frequency-domain information via fast-fourier transform (FFT). Because LLM activations are indexed by token order rather than physical time, we imposed an explicit 4 Hz clock and sampled at 4 Hz, which by Nyquist confines the analyzable band to 0–2 Hz. This explicit time axis mirrors prior continuous-time encoding approaches that reconstruct model features on a continuous timeline before comparing them with neural data [Jain et al., 2020], thereby enabling temporally interpretable model–brain correspondence. This adjustment ensured that the syntactic rhythms analogous to those observed in human brain data could be captured within the model activations.

LLMs, with their multiple layers and thousands of MLP neurons per layer, require a systematic approach to detect which neurons are responsible for either sentence or phrase processing. HFTP introduces a unified probe to detect significant syntactic processing units, applicable to both LLMs and human brain data. For the LLMs, we conducted a permutation test, randomizing the model activations derived from the structured input corpus 1000 times. The original frequency bins at 1 Hz and 2 Hz, representing sentence and phrase rhythms respectively (their real parts of amplitudes are denoted as $\text{real}[\text{amp}(1\text{ Hz})]$, $\text{real}[\text{amp}(2\text{ Hz})]$), were compared to the 95% confidence interval (CI) generated

by the distribution of permuted activations. Neurons whose $\text{real}[\text{amp}(1\text{ Hz})]$ and $\text{real}[\text{amp}(2\text{ Hz})]$ values exceeded this threshold were classified as *significant MLP neurons* (see 1), indicating their involvement in syntactic processing with statistical robustness against random noise.

**Definition 1** (Significant MLP Neurons). *For a fixed frequency $f$, a neuron is a significant MLP neuron, if and only if its FFT result satisfies*

$$\text{real}[\text{amp}(f)] \notin 95\% \text{ CI of permuted distribution.} \tag{1}$$

*The set containing all the significant MLP neurons in terms of frequency $f$ is denoted as $\mathbb{S}_f$.*

Since the *significant MLP neurons* are distributed almost uniformly across all layers, detecting the specific neurons that contribute to sentence and phrase processing requires a more objective and systematic method. We then applied z-scores to the FFT amplitudes at 1 Hz and 2 Hz in both the experimental and control groups for all *significant MLP neurons* across layers. The z-score deviation $z_f(n)$ between the experimental and control groups was then calculated for each neuron. This deviation helps minimize semantic confounds by isolating frequency-specific syntactic effects. Sentence and phrase MLP neurons were defined as those whose z-scores deviated by more than two standard deviations from the mean, at 1 Hz and 2 Hz, respectively (see 2).

**Definition 2** (Sentence MLP Neurons and Phrase MLP Neurons). *A neuron $n$ is defined as a sentence/phrase MLP neuron if it satisfies*

$$n \in \mathbb{S}_f, \quad z_f(n) \geq \mu_{z_f} + 2\sigma_{z_f}, \tag{2}$$

*where $z_f(n)$ denotes the z-score deviation of the FFT amplitude between experimental and control groups for neuron $n$ at frequency $f$, $\mu_{z_f}$ denotes the mean z-score across all neurons for the frequency $f$, $\sigma_{z_f}$ denotes the standard deviation of z-scores across all neurons for the frequency $f$, and the frequency $f$ is specified as 1 Hz and 2 Hz for sentence and phrase MLP neuron respectively.*

Following this, we identified and analysed sentence and phrase MLP neurons across layers and LLMs, with full details provided in Section 4.1. We also conducted bilingual experiments to assess the ability of different LLMs to perceive syntactic structures across Chinese and English (see Appendix C).

### 3.3 Syntactic structure probe in the human brain

We recorded sEEG from 26 native Chinese speakers while they listened to two Chinese auditory corpora. In the **sentence** corpus, nine four-syllable sentences were concatenated per trial; in the **phrase** corpus, eighteen two-syllable phrases were concatenated. Each corpus comprised 40 trials per subject. Syllables were 250 ms long, and signals were sampled at 512 Hz (2048 Hz for one participant). To minimize onset-related responses, subsequent analyses used only the final 32 syllables of each trial.

To analyze the sEEG data, we employed inter-trial phase coherence (ITPC), a frequency-domain method relatively resistant to noise that quantifies the consistency of phase relationships in oscillatory brain activity across multiple trials [Cohen, 2014]. sEEG channel localization was performed similarly to previous studies [Xu et al., 2023, Wang et al., 2024]; all channels were mapped to brain regions defined by the Automated Anatomical Labeling (AAL) system. We then grouped certain AAL regions to form 12 brain regions of interest (ROIs) (details in Appendix I). Subsequent experiments were conducted based on brain ROIs.

As previously outlined, the HFTP proposes a unified syntactic structure probe for both LLMs and human brain data. For the human brain analysis, we employed the same permutation testing procedure on the time-domain sEEG data that captured cortical activity during listening to Chinese corpora. Specifically, ITPC results were randomized 1000 times for each channel in each subject. The original frequency bins, $\text{real}[\text{amp}(1\text{ Hz})]$ and $\text{real}[\text{amp}(2\text{ Hz})]$, were then assessed to determine whether they fell within the 95% confidence interval of the permuted ITPC distribution (see 3).

**Definition 3** (Sentence channels and Phrase channels). *A channel $c$ is defined as a sentence/phrase channel if its ITPC result satisfies*

$$\text{real}[\text{amp}(f)] \notin 95\% \text{ CI of permuted ITPC,} \tag{3}$$

*where $f = 1$ Hz for sentence channel and $f = 2$ Hz for phrase channel.*

Using this probe, we identified and analyzed the distribution of sentence and phrase channels across various brain ROIs, with full details provided in Section 4.2.

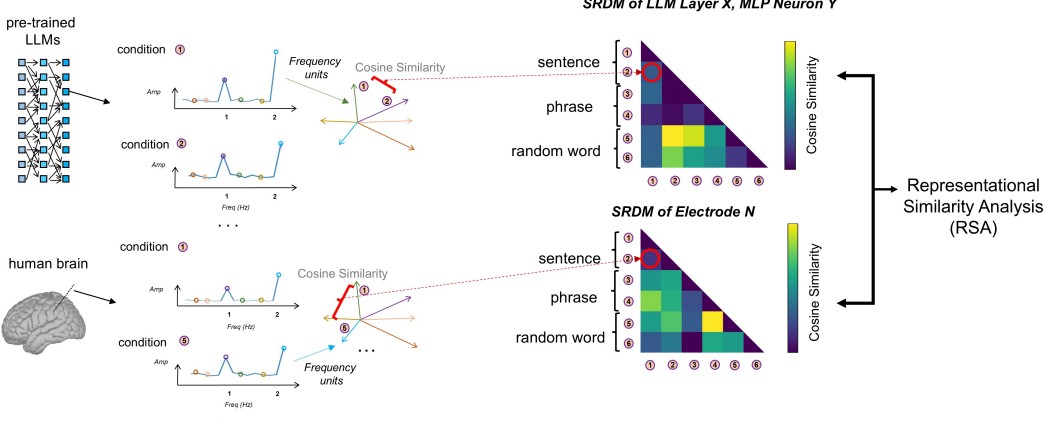

Figure 2: Alignment pipeline between LLMs and the human brain. SRDMs are computed for exclusive sentence/phrase and sentence&phrase MLP neurons and brain channels by comparing cosine similarities across different conditions. Subsequently, RSA (using Spearman correlation) is applied to quantify the similarity between the two SRDMs, thereby assessing the correspondence between model and brain representations.

## 3.4 Alignment of syntactic structure representations of LLMs with the human brain

To explore syntactic structure representation alignment between LLMs and the human brain, we compared their frequency-domain representations using the same **sentence** and **phrase** corpora as in the sEEG experiment. For each computational unit (a MLP neuron in LLMs or an sEEG channel in the human brain), we extracted amplitude in the frequency spectrum as the feature vector. This approach creates a multi-dimensional space based on frequency-domain features, where each syntactic structure corresponds to a specific point in this space (see Figure 2). We then computed the distances between these points for different syntactic structures within the same computational unit using cosine similarity. Through pairwise comparisons, we constructed Structure Representational Dissimilarity Matrices (SRDMs) for each computational unit, which are similar to Representational Dissimilarity Matrices (RDMs) but specifically capture the representations of syntactic structures [Cichy et al., 2014, Khaligh-Razavi and Kriegeskorte, 2014]. We then applied Representational Similarity Analysis (RSA) to enable cross-modal comparisons between LLMs and brain data, correlating the representations in both systems [Kriegeskorte et al., 2008]. This approach quantified alignment and used statistical tests to detect significant overlaps. We introduced two measures: model-brain similarity $S(m, b)$ and model-region similarity $S(m, b_r)$, to evaluate alignment globally and in specific brain ROI. We also used the contribution ratio $CR_r$ to assess the impact of each region on the alignment (see Appendix D). For more details on the alignment pipeline, see Appendix A. The comprehensive discussion of the alignment results can be found in Section 4.3. Additionally, as an orthogonal check, we implemented a predictive encoding analysis (see Appendix B) to assess alignment independently of SRDM-RSA.

## 4 Experiments

We used HFTP to assess syntactic processing in the human brain and LLMs, and aligned their frequency-domain representations to evaluate their similarity.

### 4.1 MLP neurons represent sentences and phrases in LLMs

Using the HFTP method, we identified neurons in all six models that selectively represent sentences (sentence MLP neurons), phrases (phrase MLP neurons), and neurons that simultaneously represent both (sentence & phrase MLP neurons). In Figure 3, we highlight representative MLP neurons exhibiting four hierarchical frequency patterns: a significant peak at the sentence frequency ($f_{\text{sentence}} = 1$ Hz), a significant peak at the phrase frequency ($f_{\text{phrase}} = 2$ Hz), dual peaks at both

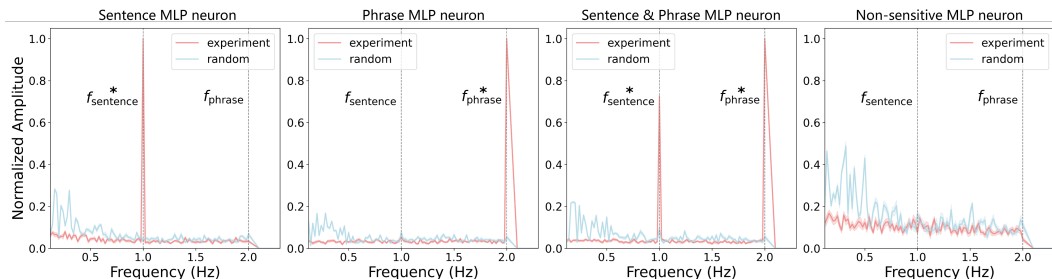

Figure 3: Hierarchical frequency patterns of MLP neurons selectively represent sentence features, phrase features, shared features of both and non-sensitive feature (from left to the right). Here, "experiment" denotes the original corpus, while "random" indicates the randomized version. Shaded bands show $\pm 1$ s.e.m. computed across 10 shuffled-input activation partitions. Significant peaks (*$p < 0.05$, FDR corrected) indicate amplitudes stronger than neighboring frequencies within $\pm 0.5$ Hz. "Normalized Amplitude" represents the curves and bands scaled to a range of 0 to 1.

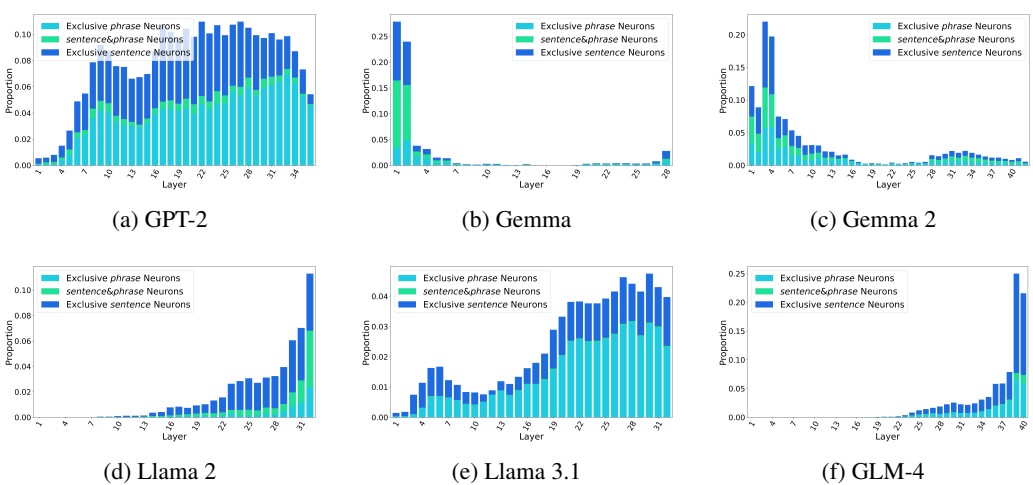

Figure 4: Statistics of exclusive sentence/phrase MLP neurons and sentence & phrase MLP neurons in each layer across six LLMs.

$f_{\text{sentence}}$ and $f_{\text{phrase}}$, and no significant peaks. Frequencies beyond 2 Hz have been artificially set to zero for smoothness in the representation.

Figure 4 shows the distribution of exclusive sentence/phrase neurons and sentence & phrase across different layers, based on experiments using the Chinese syntactic corpus (see Table 6). All the models contain neurons dedicated to capturing sentences and phrases, demonstrating their ability to encode the syntactic hierarchies of human language. However, distinct distribution patterns suggest varied syntactic processing strategies: Llama and GLM primarily process syntactic information in later layers, reflecting a more integrated approach. GPT, on the other hand, exhibits higher concentrations of sentence and phrase MLP neurons in its middle layers, suggesting a balanced intermediate strategy. In contrast, Gemma demonstrates a distinct preference for dense concentrations of neurons in the early layers, indicating an emphasis on initial-stage syntactic processing. These divergences may reflect where hierarchical composition occurs: models that infer constituent structure from short-range cues may compute sentence/phrase integration early and propagate it via residual paths, whereas models that rely on broader context may defer integration to later layers. Training dynamics may further partition depth into composition versus context-mixing stages, reallocating syntactic signals across layers even under identical inputs.

A comparative analysis reveals a significant decrease in the maximum proportions of all syntactic MLP neurons in the Llama and Gemma models, dropping from 11% in Llama 2 to 4.5% in Llama 3.1, and from 27% in Gemma to 22% in Gemma 2. As Llama 3.1 and Gemma 2 are updated versions of Llama 2 and Gemma, respectively, this trend suggests a potential shift in computational resource

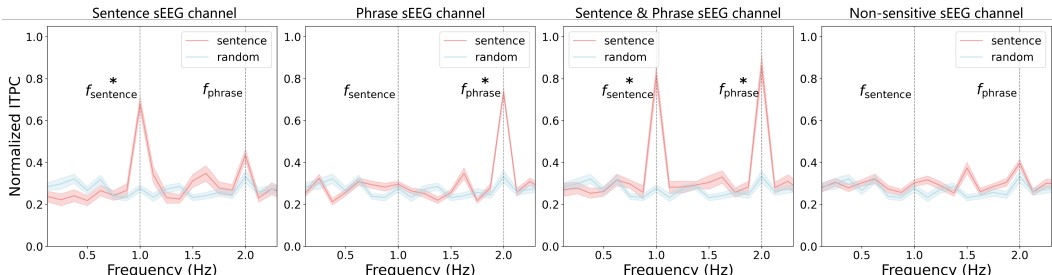

Figure 5: Hierarchical frequency patterns of sEEG channels selectively represent sentence features, phrase features, shared features of both and non-sensitive feature (from left to the right). Here, "setence" denotes the original **sentence** corpus, while "random" indicates the randomized version. Shaded bands show $\pm 1$ s.e.m. computed across channels. Significant peaks (*$p < 0.05$, FDR corrected) indicate amplitudes stronger than neighboring frequencies within $\pm 0.5$ Hz. "Normalized ITPC" represents the curves and bands scaled to a range of 0 to 1.

allocation. To enhance performance on complex tasks, Llama 3.1 and Gemma 2 may reduce their specialized processing of syntactic structures (sentence and phrase), reallocating neurons to support these advanced capabilities.

Additionally, correlations between sentence- and phrase-selective MLP neurons across layers were observed in all six models, with high coefficients (GPT-2 ($r = 0.754$), Gemma ($r = 0.994$), Gemma 2 ($r = 0.994$), Llama 2 ($r = 0.912$), Llama 3.1 ($r = 0.886$), GLM ($r = 0.993$)). This evidence indicates that the same layers are co-recruited for phrase- and sentence-level processing, consistent with a shared compositional mechanism rather than segregated pathways.

## 4.2 Sentences and phrases representations in the human brain

Using the syntactic structure probe, we identified sEEG channels that selectively represent sentence- and phrase-level structure in the **sentence** corpus. Each channel reflects collective responses from nearby neuronal populations, providing high temporal resolution and sufficient spatial specificity. As shown in Figure 5, we found channels representing sentence and phrase, as well as channels with shared representations, alongside channels with no selectivity. These patterns parallel those in LLMs, indicating that HFTP effectively interrogates internal syntactic representations in both systems.

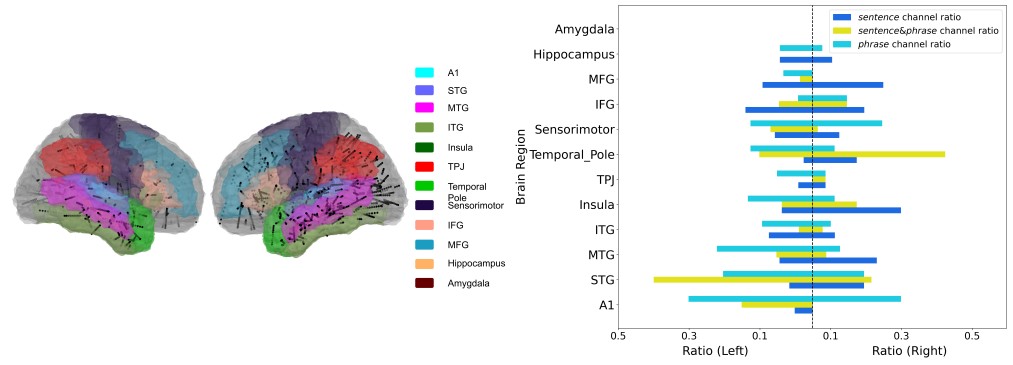

(a) sEEG channel locations and Brain ROIs      (b) Significant sEEG channel distribution

Figure 6: (a) Brain ROIs of the left and right hemispheres used in this study. The black markers represent the sEEG channel locations across all participants. (b) Distribution of significant exclusive sentence/phrase and sentence & phrase channels (**sentence** corpus) in different brain ROIs.

Analogous to our analysis of the distribution of syntactic MLP neurons across LLM layers, we computed the proportions of sentence- and phrase-selective channels within each brain ROI. As shown in Figure 6, phrase channels decrease from lower layers (A1) to higher layers (e.g., IFG), while sentence channels show the opposite trend, increasing at higher brain layers. This pattern

Table 1: Averaged top-100 Spearman correlations between sEEG-channel SRDMs and MLP-neuron SRDMs for the **sentence** corpus, separated by left (L) and right (R) hemispheres. '/' indicates that no channels from that hemisphere appear in the top-100 set. The first data row reports the overall model–brain similarity $S(m, b)$; each subsequent brain ROI row reports the model–region similarity $S(m, b_r)$. Bold in the $S(m, b)$ row flags the model with the highest overall alignment, while bold entries within each brain ROI row mark the three regions most strongly correlated for that model.

| | GPT-2 | | Gemma | | Gemma 2 | | Llama 2 | | Llama 3.1 | | GLM-4 | |
|---|---|---|---|---|---|---|---|---|---|---|---|---|
| | L | R | L | R | L | R | L | R | L | R | L | R |
| $S(m, b)$ | **0.654** | 0.442 | 0.582 | 0.411 | 0.644 | 0.450 | 0.645 | 0.439 | 0.514 | 0.405 | 0.630 | 0.445 |
| A1 | **0.683** | 0.423 | **0.642** | 0.358 | **0.702** | 0.333 | 0.649 | 0.547 | 0.514 | 0.403 | **0.664** | 0.374 |
| STG | 0.667 | 0.422 | **0.593** | 0.386 | 0.654 | 0.410 | **0.672** | 0.453 | 0.507 | 0.392 | **0.647** | 0.412 |
| MTG | 0.674 | 0.392 | 0.584 | 0.383 | **0.659** | 0.411 | **0.674** | 0.409 | **0.521** | 0.408 | 0.645 | 0.397 |
| ITG | 0.637 | 0.444 | 0.578 | 0.406 | 0.631 | 0.448 | 0.629 | 0.426 | 0.509 | 0.401 | 0.615 | 0.439 |
| Insula | 0.624 | 0.460 | 0.551 | 0.425 | 0.600 | 0.476 | 0.630 | 0.446 | **0.518** | 0.422 | 0.604 | 0.475 |
| TPJ | 0.610 | 0.452 | 0.566 | 0.373 | 0.641 | 0.410 | 0.619 | 0.400 | **0.518** | 0.408 | 0.606 | 0.438 |
| Temporal Pole | 0.648 | 0.473 | 0.556 | 0.470 | 0.643 | 0.558 | 0.610 | 0.469 | 0.494 | 0.448 | 0.616 | 0.483 |
| Sensorimotor | 0.637 | 0.462 | 0.567 | 0.426 | 0.622 | 0.448 | 0.624 | 0.446 | 0.505 | 0.396 | 0.617 | 0.463 |
| IFG | **0.694** | 0.463 | **0.603** | 0.466 | **0.670** | 0.496 | **0.665** | 0.491 | 0.513 | 0.410 | **0.646** | 0.490 |
| MFG | 0.615 | 0.436 | 0.557 | 0.401 | 0.585 | 0.489 | 0.597 | 0.367 | 0.510 | 0.397 | 0.588 | 0.473 |
| Hippocampus | **0.698** | 0.405 | 0.553 | 0.408 | 0.626 | 0.428 | 0.657 | 0.413 | 0.534 | 0.390 | 0.613 | 0.434 |
| Amygdala | / | 0.489 | 0.566 | 0.454 | / | 0.472 | / | 0.558 | 0.496 | 0.377 | / | 0.508 |

aligns with earlier MEG studies Sheng et al. [2019], supporting distinct processing mechanisms for sentences and phrases. Correlations between sentence and phrase channels across brain ROIs in both hemispheres revealed no significant relationship (left: $r$ = -0.169, $p$ = 0.870; right: $r$ = -0.197, $p$ = 0.539), suggesting that sentence and phrase processing operate independently. This contrasts with the behavior of LLMs, implying that while the human brain segregates sentence and phrase processing across different regions, LLMs integrate both syntactic levels within the same model layers. This highlights that the layered representations of LLMs may not align directly with the distinct processing roles observed in brain ROIs.

### 4.3 Alignment of syntactic structure representations between LLMs and the human brain

As the layered representations of LLMs do not correspond directly to the distinct processing functions of different brain ROIs, we sought to investigate whether overall syntactic structure representations in LLMs are comparable to those in the human brain, both globally and across individual brain ROIs. To accomplish this, we used representational alignment to quantify correspondence between model and brain frequency-domain features. Detailed procedures are provided in Appendix A.

The alignment results across both **sentence** and **phrase** corpora revealed consistent patterns of model-brain correspondence. As shown in Tables 1 and 2, we observed that the similarity between LLMs and the left hemisphere is notably higher than that with the right hemisphere across both structure levels. Brain-wise one-way ANOVAs confirmed significant model differences for sentence-level ($F$ = 59.74, $\eta^2$ = 0.027, $p$ < 0.001) and phrase-level processing ($F$ = 12.10, $\eta^2$ = 0.018, $p$ < 0.001). Region-wise ANOVAs (FDR-corrected) further showed a strongly left-lateralized effect across corpora, concentrated in core language cortex (e.g., STG, MTG, IFG).

Examining individual model performance, GPT-2 exhibited the highest average correlation with human brain activity across both sentence ($S(m, b)$ = 0.654, denoted $S$; $L$) and phrase levels ($S$ = 0.654, $L$). Gemma 2 ($S$ = 0.644, $L$ for **sentence**; $S$ = 0.628, $L$ for **phrase**) consistently outperformed Gemma ($S$ = 0.582, $L$ for **sentence**; $S$ = 0.575, $L$ for **phrase**) at both corpora, attributed to architectural improvements [Team et al., 2024a]. Most notably, Llama 3.1 ($S$ = 0.514, $L$ for **sentence**; $S$ = 0.522, $L$ for **phrase**) showed consistently lower alignment than Llama 2 ($S$ = 0.645, $L$ for **sentence**; $S$ = 0.648, $L$ for **phrase**) across both processing levels. This counterintuitive pattern is explained at the pretraining level: Llama 3.1 was trained on a substantially larger corpus emphasizing code, reasoning, and multilingual text, which dilutes language-specific regularities. Additionally, extensive reliance on synthetic data for capability-targeted curation introduces distributional shifts away from naturalistic language statistics [Dubey et al., 2024, Touvron et al., 2023]. These findings echo evidence

Table 2: Averaged top-100 Spearman correlations between sEEG-channel SRDMs and MLP-neuron SRDMs for the **phrase** corpus, separated by left (L) and right (R) hemispheres. '/' indicates that no channels from that hemisphere appear in the top-100 set. The first data row reports the overall model–brain similarity $S(m, b)$; each subsequent brain ROI row reports the model–region similarity $S(m, b_r)$. Bold in the $S(m, b)$ row flags the model with the highest overall alignment, while bold entries within each brain ROI row mark the three regions most strongly correlated for that model.

| | GPT-2 | | Gemma | | Gemma 2 | | Llama 2 | | Llama 3.1 | | GLM-4 | |
|---|---|---|---|---|---|---|---|---|---|---|---|---|
| | L | R | L | R | L | R | L | R | L | R | L | R |
| $S(m, b)$ | **0.654** | 0.441 | 0.575 | 0.416 | 0.628 | 0.443 | 0.648 | 0.435 | 0.522 | 0.404 | 0.626 | 0.437 |
| A1 | **0.669** | 0.420 | **0.610** | 0.315 | **0.665** | 0.326 | 0.628 | 0.576 | **0.526** | 0.372 | 0.627 | 0.374 |
| STG | 0.665 | 0.415 | **0.591** | 0.388 | 0.643 | 0.395 | **0.674** | 0.447 | **0.530** | 0.402 | **0.649** | 0.401 |
| MTG | 0.665 | 0.391 | 0.564 | 0.408 | **0.645** | 0.415 | **0.675** | 0.415 | 0.521 | 0.401 | **0.641** | 0.408 |
| ITG | 0.637 | 0.447 | 0.558 | 0.421 | 0.611 | 0.440 | 0.632 | 0.421 | 0.518 | 0.398 | 0.621 | 0.427 |
| Insula | 0.627 | 0.468 | 0.556 | 0.433 | 0.596 | 0.465 | 0.632 | 0.453 | 0.519 | 0.409 | 0.602 | 0.470 |
| TPJ | 0.616 | 0.442 | 0.563 | 0.405 | 0.608 | 0.423 | 0.615 | 0.384 | **0.542** | 0.399 | 0.588 | 0.433 |
| Temporal Pole | **0.680** | 0.453 | 0.555 | 0.394 | 0.626 | 0.501 | 0.623 | 0.456 | 0.505 | 0.450 | 0.583 | 0.442 |
| Sensorimotor | 0.645 | 0.475 | 0.565 | 0.443 | 0.607 | 0.449 | 0.629 | 0.442 | 0.511 | 0.391 | 0.613 | 0.460 |
| IFG | **0.700** | 0.450 | **0.635** | 0.424 | **0.656** | 0.491 | **0.674** | 0.488 | 0.514 | 0.436 | **0.636** | 0.480 |
| MFG | 0.618 | 0.409 | 0.562 | 0.421 | 0.583 | 0.500 | 0.601 | 0.365 | 0.505 | 0.379 | 0.577 | 0.475 |
| Hippocampus | 0.656 | 0.398 | 0.541 | 0.418 | 0.602 | 0.432 | 0.640 | 0.398 | 0.516 | 0.406 | 0.566 | 0.434 |
| Amygdala | / | 0.483 | / | 0.433 | / | 0.475 | / | 0.552 | 0.468 | 0.386 | / | 0.492 |

that scaling alone fails to secure robust predicate–argument structure—especially for long-range, boundary-sensitive roles—revealing persistent human–model structural gaps [Cheng et al., 2024].

Additionally, the key brain ROIs for each model, which are primarily located in the left hemisphere, highlighted regions that are critical for syntactic processing at both levels of the syntactic hierarchy. These regions include the left A1, STG, MTG, and IFG, with many of the LLMs exhibiting particularly strong correlations in these areas across both structure, emphasizing their role in syntactic functions. These converging findings reinforce the robustness of the HFTP approach and suggest its potential as a valuable tool for future studies of model-brain alignment.

# 5   Conclusion

This study advances syntactic processing by introducing the Hierarchical Frequency Tagging Probe (HFTP), a unified framework for dissecting neuron-wise sentence and phrase representations in LLMs, population-level patterns in the human brain, and generalizing seamlessly to naturalistic text. The results reveal that while LLMs exhibit hierarchical syntactic processing and alignment with left-hemisphere brain activity, the mechanisms underlying their representations diverge significantly from those in human cortical regions. Notably, newer models like Gemma 2 demonstrate improved alignment, whereas others, such as Llama 3.1, show weaker human-model correlations despite enhanced task performance. These findings underscore the need to refine LLM architectures for more human-like syntactic processing and establish HFTP as a bridge between computational linguistics and cognitive neuroscience. Finally, the societal implications of this work are two-sided: positively, HFTP can support safer, more controllable models and inform non-invasive diagnostics via spectral markers; negatively, the same interpretability could be misused to optimize persuasive manipulation and, if linked with personal neural data, undermine privacy.

# 6   Limitation

Although we applied HFTP to both Chinese and English corpora for LLMs, our sEEG data were collected in China and primarily involve Chinese stimuli. We have preliminary English responses from a single Chinese native participant (see Appendix F), but deeper cross-linguistic analyses are pending; future work will recruit native English speakers to validate alignment on English and enable rigorous cross-linguistic tests. Additionally, we evaluated only a small set of model architectures and parameter scales, so the key universal mechanisms driving model–brain syntactic alignment remain unclear and warrant further investigation.

## Acknowledgments and Disclosure of Funding

This work was supported by the National Natural Science Foundation of China (32441106, 32171039, T2421004) and National Science and Technology Innovation 2030 Major Program (2022ZD0204804, 2022ZD0204802).

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

# NeurIPS Paper Checklist

1. **Claims**

   Question: Do the main claims made in the abstract and introduction accurately reflect the paper's contributions and scope?

   Answer: [Yes]

   Justification: The Abstract and Introduction sections explicitly enumerate the HFTP framework, its application to both LLMs and brain data, and summarize the key empirical and methodological contributions, accurately reflecting the paper's scope.

   Guidelines:

   - The answer NA means that the abstract and introduction do not include the claims made in the paper.
   - The abstract and/or introduction should clearly state the claims made, including the contributions made in the paper and important assumptions and limitations. A No or NA answer to this question will not be perceived well by the reviewers.
   - The claims made should match theoretical and experimental results, and reflect how much the results can be expected to generalize to other settings.
   - It is fine to include aspirational goals as motivation as long as it is clear that these goals are not attained by the paper.

2. **Limitations**

   Question: Does the paper discuss the limitations of the work performed by the authors?

   Answer: [Yes]

   Justification: In Section 6, We explicitly acknowledge two limitations: (i) our sEEG data are restricted to Chinese stimuli, so future work must include native-English recordings to test cross-language generalization; and (ii) we evaluate only six LLM architectures and sizes, leaving the core drivers of model–brain syntactic alignment for further study.

   Guidelines:

   - The answer NA means that the paper has no limitation while the answer No means that the paper has limitations, but those are not discussed in the paper.
   - The authors are encouraged to create a separate "Limitations" section in their paper.
   - The paper should point out any strong assumptions and how robust the results are to violations of these assumptions (e.g., independence assumptions, noiseless settings, model well-specification, asymptotic approximations only holding locally). The authors should reflect on how these assumptions might be violated in practice and what the implications would be.
   - The authors should reflect on the scope of the claims made, e.g., if the approach was only tested on a few datasets or with a few runs. In general, empirical results often depend on implicit assumptions, which should be articulated.
   - The authors should reflect on the factors that influence the performance of the approach. For example, a facial recognition algorithm may perform poorly when image resolution is low or images are taken in low lighting. Or a speech-to-text system might not be used reliably to provide closed captions for online lectures because it fails to handle technical jargon.
   - The authors should discuss the computational efficiency of the proposed algorithms and how they scale with dataset size.
   - If applicable, the authors should discuss possible limitations of their approach to address problems of privacy and fairness.
   - While the authors might fear that complete honesty about limitations might be used by reviewers as grounds for rejection, a worse outcome might be that reviewers discover limitations that aren't acknowledged in the paper. The authors should use their best judgment and recognize that individual actions in favor of transparency play an important role in developing norms that preserve the integrity of the community. Reviewers will be specifically instructed to not penalize honesty concerning limitations.

3. **Theory assumptions and proofs**

Question: For each theoretical result, does the paper provide the full set of assumptions and a complete (and correct) proof?

Answer: [NA]

Justification: This work is empirical and method-driven—defining how to compute sentence and phrase unit responses, layer-wise alignment similarity, and contribution ratios—but does not present formal theorems that require assumption lists or mathematical proofs.

Guidelines:

- The answer NA means that the paper does not include theoretical results.
- All the theorems, formulas, and proofs in the paper should be numbered and cross-referenced.
- All assumptions should be clearly stated or referenced in the statement of any theorems.
- The proofs can either appear in the main paper or the supplemental material, but if they appear in the supplemental material, the authors are encouraged to provide a short proof sketch to provide intuition.
- Inversely, any informal proof provided in the core of the paper should be complemented by formal proofs provided in appendix or supplemental material.
- Theorems and Lemmas that the proof relies upon should be properly referenced.

4. **Experimental result reproducibility**

Question: Does the paper fully disclose all the information needed to reproduce the main experimental results of the paper to the extent that it affects the main claims and/or conclusions of the paper (regardless of whether the code and data are provided or not)?

Answer: [Yes]

Justification: We give end-to-end details of every experiment—how to detect sentence and phrase representations in LLMs (Section 3.2), how to identify corresponding signals in human sEEG data (Section 3.3), and how to compute layer-wise model–brain alignment (Section 3.4 and Appendix A). These sections include all parameters, sampling-rate choices, FFT settings, permutation tests, and alignment-pipeline steps needed to reproduce the analyses.

Guidelines:

- The answer NA means that the paper does not include experiments.
- If the paper includes experiments, a No answer to this question will not be perceived well by the reviewers: Making the paper reproducible is important, regardless of whether the code and data are provided or not.
- If the contribution is a dataset and/or model, the authors should describe the steps taken to make their results reproducible or verifiable.
- Depending on the contribution, reproducibility can be accomplished in various ways. For example, if the contribution is a novel architecture, describing the architecture fully might suffice, or if the contribution is a specific model and empirical evaluation, it may be necessary to either make it possible for others to replicate the model with the same dataset, or provide access to the model. In general. releasing code and data is often one good way to accomplish this, but reproducibility can also be provided via detailed instructions for how to replicate the results, access to a hosted model (e.g., in the case of a large language model), releasing of a model checkpoint, or other means that are appropriate to the research performed.
- While NeurIPS does not require releasing code, the conference does require all submissions to provide some reasonable avenue for reproducibility, which may depend on the nature of the contribution. For example
  - (a) If the contribution is primarily a new algorithm, the paper should make it clear how to reproduce that algorithm.
  - (b) If the contribution is primarily a new model architecture, the paper should describe the architecture clearly and fully.
  - (c) If the contribution is a new model (e.g., a large language model), then there should either be a way to access this model for reproducing the results or a way to reproduce the model (e.g., with an open-source dataset or instructions for how to construct the dataset).

(d) We recognize that reproducibility may be tricky in some cases, in which case authors are welcome to describe the particular way they provide for reproducibility. In the case of closed-source models, it may be that access to the model is limited in some way (e.g., to registered users), but it should be possible for other researchers to have some path to reproducing or verifying the results.

5. **Open access to data and code**

    Question: Does the paper provide open access to the data and code, with sufficient instructions to faithfully reproduce the main experimental results, as described in supplemental material?

    Answer: [Yes]

    Justification:We provide all corpora used for LLM experiments—Chinese and English four-syllable/word corpora and naturalistic texts—in Section H. For our sEEG and brain–LLM alignment studies, we adopted the sentence/phrase corpus from [Sheng et al., 2019], which follows the exact structural format of the four-syllable Chinese corpus, with only semantic content differing. This project is available at `https://github.com/LilTiger/HFTP`..

    Guidelines:

    - The answer NA means that paper does not include experiments requiring code.
    - Please see the NeurIPS code and data submission guidelines (`https://nips.cc/public/guides/CodeSubmissionPolicy`) for more details.
    - While we encourage the release of code and data, we understand that this might not be possible, so "No" is an acceptable answer. Papers cannot be rejected simply for not including code, unless this is central to the contribution (e.g., for a new open-source benchmark).
    - The instructions should contain the exact command and environment needed to run to reproduce the results. See the NeurIPS code and data submission guidelines (`https://nips.cc/public/guides/CodeSubmissionPolicy`) for more details.
    - The authors should provide instructions on data access and preparation, including how to access the raw data, preprocessed data, intermediate data, and generated data, etc.
    - The authors should provide scripts to reproduce all experimental results for the new proposed method and baselines. If only a subset of experiments are reproducible, they should state which ones are omitted from the script and why.
    - At submission time, to preserve anonymity, the authors should release anonymized versions (if applicable).
    - Providing as much information as possible in supplemental material (appended to the paper) is recommended, but including URLs to data and code is permitted.

6. **Experimental setting/details**

    Question: Does the paper specify all the training and test details (e.g., data splits, hyper-parameters, how they were chosen, type of optimizer, etc.) necessary to understand the results?

    Answer: [Yes]

    Justification: We give full details of the LLM evaluation—concatenating each Chinese/English corpus sentence stream and extracting MLP-layer activations in inference mode—in Section 3.2, and describe the human sEEG passive-listening paradigm with fixed 512/2048 Hz audio stimuli in Section 3.3.

    Guidelines:

    - The answer NA means that the paper does not include experiments.
    - The experimental setting should be presented in the core of the paper to a level of detail that is necessary to appreciate the results and make sense of them.
    - The full details can be provided either with the code, in appendix, or as supplemental material.

7. **Experiment statistical significance**

    Question: Does the paper report error bars suitably and correctly defined or other appropriate information about the statistical significance of the experiments?

Answer: [Yes]

Justification: We used paired $t$-tests in Figures 3 and 5 to confirm that power at hierarchical frequencies is significantly higher in the experimental than in the control conditions, and we report Pearson correlations in Sections 4.1 and 4.2 to assess covariation across layers and ROIs, as well as one-way ANOVAs to test model alignment effects (Sections 4.3, B); all $p$-values are FDR-corrected for multiple comparisons. Additionally, we employ permutation tests for both the LLM and sEEG probes (Sections 3.2, 3.3).

Guidelines:

- The answer NA means that the paper does not include experiments.
- The authors should answer "Yes" if the results are accompanied by error bars, confidence intervals, or statistical significance tests, at least for the experiments that support the main claims of the paper.
- The factors of variability that the error bars are capturing should be clearly stated (for example, train/test split, initialization, random drawing of some parameter, or overall run with given experimental conditions).
- The method for calculating the error bars should be explained (closed form formula, call to a library function, bootstrap, etc.)
- The assumptions made should be given (e.g., Normally distributed errors).
- It should be clear whether the error bar is the standard deviation or the standard error of the mean.
- It is OK to report 1-sigma error bars, but one should state it. The authors should preferably report a 2-sigma error bar than state that they have a 96% CI, if the hypothesis of Normality of errors is not verified.
- For asymmetric distributions, the authors should be careful not to show in tables or figures symmetric error bars that would yield results that are out of range (e.g. negative error rates).
- If error bars are reported in tables or plots, The authors should explain in the text how they were calculated and reference the corresponding figures or tables in the text.

8. **Experiments compute resources**

Question: For each experiment, does the paper provide sufficient information on the computer resources (type of compute workers, memory, time of execution) needed to reproduce the experiments?

Answer: [Yes]

Justification: We ran all LLM inference and permutation tests on a server with AMD EPYC 7282 (16 cores, 32 threads, 256 GB RAM) and two NVIDIA A100-40 GB GPUs using bfloat16 precision. The permutation tests used to probe sentence/phrase MLP neurons require several hours on a single GPU, whereas all other representational analyses complete in under an hour with minimal additional resources.

Guidelines:

- The answer NA means that the paper does not include experiments.
- The paper should indicate the type of compute workers CPU or GPU, internal cluster, or cloud provider, including relevant memory and storage.
- The paper should provide the amount of compute required for each of the individual experimental runs as well as estimate the total compute.
- The paper should disclose whether the full research project required more compute than the experiments reported in the paper (e.g., preliminary or failed experiments that didn't make it into the paper).

9. **Code of ethics**

Question: Does the research conducted in the paper conform, in every respect, with the NeurIPS Code of Ethics https://neurips.cc/public/EthicsGuidelines?

Answer: [Yes]

Justification: All human data were collected under IRB-approved protocols with informed consent, no personal identifiers are used, and our computational methods present no foreseeable harm.

Guidelines:

- The answer NA means that the authors have not reviewed the NeurIPS Code of Ethics.
- If the authors answer No, they should explain the special circumstances that require a deviation from the Code of Ethics.
- The authors should make sure to preserve anonymity (e.g., if there is a special consideration due to laws or regulations in their jurisdiction).

10. **Broader impacts**

Question: Does the paper discuss both potential positive societal impacts and negative societal impacts of the work performed?

Answer: [Yes]

Justification: This work offers a transparent, neuro-inspired lens on LLM internals. Positive impacts include safer, more controllable models—by revealing which neurons drive hierarchical syntax we can design targeted safeguards and debiasing interventions; improved language-disorder diagnostics—spectral markers could serve as non-invasive probes of syntactic deficits; and richer human-AI interfaces that align model parsing with human cognition. Negative impacts also merit consideration: the same interpretability could be weaponised to optimise persuasive text or amplify covert influence; detailed brain-model correspondences might facilitate adversarial "neuro-phishing" that mimics neural rhythms; and, if paired with personal neural data, fine-grained model maps could erode privacy.

Guidelines:

- The answer NA means that there is no societal impact of the work performed.
- If the authors answer NA or No, they should explain why their work has no societal impact or why the paper does not address societal impact.
- Examples of negative societal impacts include potential malicious or unintended uses (e.g., disinformation, generating fake profiles, surveillance), fairness considerations (e.g., deployment of technologies that could make decisions that unfairly impact specific groups), privacy considerations, and security considerations.
- The conference expects that many papers will be foundational research and not tied to particular applications, let alone deployments. However, if there is a direct path to any negative applications, the authors should point it out. For example, it is legitimate to point out that an improvement in the quality of generative models could be used to generate deepfakes for disinformation. On the other hand, it is not needed to point out that a generic algorithm for optimizing neural networks could enable people to train models that generate Deepfakes faster.
- The authors should consider possible harms that could arise when the technology is being used as intended and functioning correctly, harms that could arise when the technology is being used as intended but gives incorrect results, and harms following from (intentional or unintentional) misuse of the technology.
- If there are negative societal impacts, the authors could also discuss possible mitigation strategies (e.g., gated release of models, providing defenses in addition to attacks, mechanisms for monitoring misuse, mechanisms to monitor how a system learns from feedback over time, improving the efficiency and accessibility of ML).

11. **Safeguards**

Question: Does the paper describe safeguards that have been put in place for responsible release of data or models that have a high risk for misuse (e.g., pretrained language models, image generators, or scraped datasets)?

Answer: [NA]

Justification: We do not release any externally scraped or dual-use models or datasets. Our study uses (i) in-house sEEG recordings collected under institutional ethical approval with informed consent, and (ii) analyses of existing, publicly available language-model checkpoints (e.g. GPT-2, Gemma, Llama 2/3) without redistributing their weights.

Guidelines:

- The answer NA means that the paper poses no such risks.

- Released models that have a high risk for misuse or dual-use should be released with necessary safeguards to allow for controlled use of the model, for example by requiring that users adhere to usage guidelines or restrictions to access the model or implementing safety filters.
- Datasets that have been scraped from the Internet could pose safety risks. The authors should describe how they avoided releasing unsafe images.
- We recognize that providing effective safeguards is challenging, and many papers do not require this, but we encourage authors to take this into account and make a best faith effort.

12. **Licenses for existing assets**

Question: Are the creators or original owners of assets (e.g., code, data, models), used in the paper, properly credited and are the license and terms of use explicitly mentioned and properly respected?

Answer: [Yes]

Justification: We correctly cite and credit the Chinese and English syntactic corpora from [Ding et al., 2016] and [Sheng et al., 2019], and we list each LLM—GPT-2, Gemma, Gemma 2, Llama 2, Llama 3.1, and GLM-4—with version, source, and license information in Section G.

Guidelines:

- The answer NA means that the paper does not use existing assets.
- The authors should cite the original paper that produced the code package or dataset.
- The authors should state which version of the asset is used and, if possible, include a URL.
- The name of the license (e.g., CC-BY 4.0) should be included for each asset.
- For scraped data from a particular source (e.g., website), the copyright and terms of service of that source should be provided.
- If assets are released, the license, copyright information, and terms of use in the package should be provided. For popular datasets, `paperswithcode.com/datasets` has curated licenses for some datasets. Their licensing guide can help determine the license of a dataset.
- For existing datasets that are re-packaged, both the original license and the license of the derived asset (if it has changed) should be provided.
- If this information is not available online, the authors are encouraged to reach out to the asset's creators.

13. **New assets**

Question: Are new assets introduced in the paper well documented and is the documentation provided alongside the assets?

Answer: [Yes]

Justification: We introduced a diverse, curated suite of corpora for the HFTP experiments: (i) two four-syllable/word syntactic corpora (Chinese and English) adapted from [Ding et al., 2016]; (ii) four human-generated naturalistic texts in Chinese and English; and (iii) two Wikipedia-derived corpora in Chinese and English. Detailed descriptions for all non-Wikipedia corpora are provided in Section H; the Wikipedia-derived corpora are hosted on `https://github.com/LilTiger/HFTP`.

Guidelines:

- The answer NA means that the paper does not release new assets.
- Researchers should communicate the details of the dataset/code/model as part of their submissions via structured templates. This includes details about training, license, limitations, etc.
- The paper should discuss whether and how consent was obtained from people whose asset is used.
- At submission time, remember to anonymize your assets (if applicable). You can either create an anonymized URL or include an anonymized zip file.

14. **Crowdsourcing and research with human subjects**

Question: For crowdsourcing experiments and research with human subjects, does the paper include the full text of instructions given to participants and screenshots, if applicable, as well as details about compensation (if any)?

Answer: [Yes]

Justification: In our sEEG experiment participants received a brief oral instruction ("please listen carefully to the Chinese corpus") and then completed 40 trials per condition (sentence, phrase, random). Compensation was ¥200 for full participation, prorated for partial completion.

Guidelines:

- The answer NA means that the paper does not involve crowdsourcing nor research with human subjects.
- Including this information in the supplemental material is fine, but if the main contribution of the paper involves human subjects, then as much detail as possible should be included in the main paper.
- According to the NeurIPS Code of Ethics, workers involved in data collection, curation, or other labor should be paid at least the minimum wage in the country of the data collector.

15. **Institutional review board (IRB) approvals or equivalent for research with human subjects**

Question: Does the paper describe potential risks incurred by study participants, whether such risks were disclosed to the subjects, and whether Institutional Review Board (IRB) approvals (or an equivalent approval/review based on the requirements of your country or institution) were obtained?

Answer: [Yes]

Justification: The study protocol received approval from our institution's IRB, minimal auditory-stimulation risks were disclosed to participants during the consent process, and all participants provided informed consent prior to data collection.

Guidelines:

- The answer NA means that the paper does not involve crowdsourcing nor research with human subjects.
- Depending on the country in which research is conducted, IRB approval (or equivalent) may be required for any human subjects research. If you obtained IRB approval, you should clearly state this in the paper.
- We recognize that the procedures for this may vary significantly between institutions and locations, and we expect authors to adhere to the NeurIPS Code of Ethics and the guidelines for their institution.
- For initial submissions, do not include any information that would break anonymity (if applicable), such as the institution conducting the review.

16. **Declaration of LLM usage**

Question: Does the paper describe the usage of LLMs if it is an important, original, or non-standard component of the core methods in this research? Note that if the LLM is used only for writing, editing, or formatting purposes and does not impact the core methodology, scientific rigorousness, or originality of the research, declaration is not required.

Answer: [Yes]

Justification: The HFTP method is built on extraction of MLP activations from six LLMs (GPT-2, Gemma, Gemma 2, Llama 2, Llama 3.1, GLM-4). Model parameters are listed in Section G, the HFTP procedure for LLMs is detailed in Section 3.2, and the LLM–brain alignment pipeline is described in Section A.

Guidelines:

- The answer NA means that the core method development in this research does not involve LLMs as any important, original, or non-standard components.
- Please refer to our LLM policy (`https://neurips.cc/Conferences/2025/LLM`) for what should or should not be described.

# A  Alignment pipeline for syntactic processing between LLMs and the human brain

In this appendix, we provide the detailed pipeline used to align the syntactic structure representations in LLMs with those in the human brain, with a focus on detecting and comparing sentence- and phrase-level representations across both systems.

**Data and experimental setup** For system-level comparability, we presented the exact same input corpora to both systems in the alignment analyses: the **sentence** corpus (four-syllable Chinese sequences) and the **phrase** corpus (two-syllable Chinese sequences).. The word-order randomized version of each corpus was used as a control condition, as detailed in Section 3.1. Each corpus comprised 40 trials, and each trial contained 36 syllables. For SRDM calculation, the corpora were divided into six experimental conditions, each with 20 trials. To attenuate onset-locked transients, sEEG signals and model activations were extracted for the final 32 syllables of each trial. These time series were then transformed to the frequency domain via FFT, and the resulting amplitude spectra were used in the subsequent alignment analyses.

**Frequency-domain analysis and SRDM construction** For the LLMs, neuron activations were transformed using FFT to capture the frequency components of structure processing across the six conditions. From this transformation, we calculated the cosine similarity between each pair of conditions, constructing an SRDM for each MLP neuron. We then averaged the SRDM for each MLP neuron in the same layer to obtain layer-wise SRDMs. Similarly, for the human brain, we calculated the ITPC to capture frequency-domain representations for each brain channel, yielding a channel-wise SRDM..

To assess the structural alignment between LLMs and the human brain, we computed the Spearman correlation $\rho$ between the SRDM of each LLM layer and the SRDM of each brain channel. We then formed layer-level model SRDMs by averaging neuron-level cosine-similarity matrices within each layer. The top 100 most relevant brain channels for each model layer were identified based on Spearman correlation, and the overlap of sentence and phrase channels in these top 100 channels was evaluated using a chi-square test. A significant overlap indicated alignment in structural processing between LLMs and brain ROIs.

**Quantification of cross-system alignment** Two key metrics were defined to quantify the structural alignment between LLMs and the human brain. The first, model-brain similarity $S(m, b)$, represents the overall similarity of syntactic processing between an LLM $m$ and the human brain $b$. It is computed as the average Spearman correlation between the SRDM of each LLM layer and the top 100 most relevant brain channels:

$$S(m, b) = \frac{1}{M} \sum_{j=1}^{M} \frac{1}{100} \sum_{i \in \text{top}(j)}^{100} \rho(L_j, C_i), \tag{4}$$

where $M$ is the number of layers in an LLM; $L_j$ and $C_i$ denote the model layer and the brain channel indexed by $j$ and $i$, respectively; $\text{top}(j)$ denotes the indices of the top 100 channels for model layer $L_j$; and $\rho(L_j, C_i)$ denotes the Spearman correlation between the model SRDM at layer $L_j$ and the SRDM of brain channel $C_i$.

The second metric, model-region similarity $S(m, b_r)$, measures the alignment between LLMs and specific brain ROIs. This is computed by averaging the Spearman correlations over the subset of the top 100 channels that fall within a particular brain ROI:

$$S(m, b_r) = \frac{1}{M} \sum_{j=1}^{M} \frac{1}{n(j, r)} \sum_{i \in \text{top}(j) \cap \mathbb{C}_r}^{n(j,r)} \rho(L_j, C_i), \tag{5}$$

where $\mathbb{C}_r$ denotes the indices of all channels belonging to region $r$, and $n(j, r)$ is the cardinality of $\text{top}(j) \cap \mathbb{C}_r$, i.e., the number of channels in region $r$ that are also among the top 100 channels for model layer $L_j$.

Note that we have sentence, phrase, and sentence & phrase neurons for the **sentence** corpus, thus we average $S(m, b)$ and $S(m, b_r)$ across the three neuron types; and, for the **phrase** corpus, there are only phrase neurons, so no averaging is required.

## B  Predictive encoding control analysis

To provide an orthogonal test of model–brain alignment that does not depend on representational similarity analysis, we implemented a predictive encoding model that asks whether layer-wise features extracted from an LLM can predict the frequency–domain sEEG responses of individual channels. All experimental settings (e.g. **sentence/phrase** corpora, block structure) were matched to those in the RDM–RSA pipeline (Section A). We also adopt the same two measures, model–brain similarity $S(m, b)$ and model–region similarity $S(m, b_r)$, so that the predictive encoding results are directly comparable to the SRDM-RSA results reported above.

**Model features** For each model layer $L_j$ we first selected significant neurons using the same procedure as in the SRDM-RSA pipeline. Within a given block, the time courses of the significant neurons were averaged to obtain one 32-syllable sequence per layer and block (the last 32 syllables). We then applied FFT to this layer-average and retained the complex coefficients within 0.5–2 Hz. Each retained coefficient was represented by its real part and imaginary part (the canonical cosine and sine components), which we treat as two features at that frequency. With two blocks per corpus, we formed the predictor matrix $X_j \in \mathbb{R}^{N \times 2}$ by stacking samples across blocks and frequencies, where $K$ denotes the number of frequency bins in 0.5–2 Hz and $N = 2K$ reflects the two blocks times $K$ frequencies. Thus each of the $N$ samples corresponds to one (block, frequency) pair and is described by a two-dimensional feature vector containing the real and imaginary parts of the aligned model coefficient.

**Brain responses** For each sEEG channel $C_i$, we computed ITPC exactly as in the SRDM-RSA analysis, yielding a complex spectrum per block. We then aligned the frequency axes of the model and brain spectra by nearest-neighbor matching (model sampling 4 Hz; sEEG 512 Hz) within the 0.5–2 Hz band. The target for predictive modeling was the band-limited spectral profile of stimulus-locked synchrony for channel $C_i$; specifically, we concatenated the ITPC amplitudes across the two blocks and across all $K$ frequencies to obtain an observation vector $y_i \in \mathbb{R}^N$. This vector summarizes the oscillatory response of channel $C_i$ within the syntactic band while preserving block-specific structure.

**Predictive model and cross-validation** For each pair $(L_j, C_i)$ we fit a ridge regression with standardized predictors,

$$\hat{\boldsymbol{\beta}} = \arg\min_{\boldsymbol{\beta}} \left\| y_i - X_j \boldsymbol{\beta} \right\|_2^2 + \alpha \|\boldsymbol{\beta}\|_2^2, \tag{6}$$

using five random splits (train/test = 70/30). The ridge penalty $\alpha$ was chosen only on the training split via inner ridge-CV over a logarithmic grid; the fitted model was then evaluated on the held-out test data. Predictive accuracy for a split was defined as Spearman correlation $\rho$ between the predicted and observed test targets. We averaged the split-wise correlations to obtain a predictive score

$$P(L_j, C_i) \;=\; \mathbb{E}_{\text{splits}} \big[ \rho \big( \hat{y}_i^{\text{test}}, y_i^{\text{test}} \big) \big], \tag{7}$$

and repeated this procedure independently for left and right hemispheres and for both corpora. Note that feature selection is performed exclusively on the model side; the sEEG data are never used to select features, preventing circularity (double-dipping).

**Aggregation into alignment metrics** To summarize predictive alignment, we follow the exact aggregation used for SRDM-RSA pipeline and therefore reuse the definitions of $S(m, b)$ and $S(m, b_r)$ introduced in Section A. The only substitution is that SRDM correlations $\rho(L_j, C_i)$ are replaced by the predictive scores $P(L_j, C_i)$. Concretely, for each layer $L_j$ we rank channels by $P(L_j, C_i)$, take the top 100, and compute $S(m, b)$ and $S(m, b_r)$ by the same layer-averaging rules as before. For the **sentence** corpus, we compute these summaries separately for sentence, phrase and sentence & phrase neurons and then report their arithmetic mean to yield a single value per model and hemisphere. For the **phrase** corpus, the selection procedure yields only phrase-selective neurons; consequently no averaging across syntactic MLP neuron types is required. This results in one model–brain score $S(m, b)$ and one score per ROI $S(m, b_r)$ for each model in each hemisphere, directly comparable to the SRDM-RSA results while relying on predictive encoding rather than representational geometries.

Across both corpora (Tables 3 and 4), a one-way ANOVA on layer-averaged top-100 predictive scores revealed reliable between-model differences in brain-level alignment $S(m, b)$ (**sentence**:

Table 3: Predictive alignment results for the **sentence** corpus. The first row reports the overall model–brain similarity $S(m,b)$; subsequent rows report the model–region similarity $S(m,b_r)$. Bold in the $S(m,b)$ row marks the highest value(s) across models/hemispheres; within each model, bold highlights the three highest $S(m,b_r)$ entries across all ROIs and hemispheres (ties bolded).

| | GPT-2 | | Gemma | | Gemma 2 | | Llama 2 | | Llama 3.1 | | GLM-4 | |
|---|---|---|---|---|---|---|---|---|---|---|---|---|
| | L | R | L | R | L | R | L | R | L | R | L | R |
| $S(m,b)$ | **0.487** | 0.371 | 0.467 | 0.364 | 0.469 | 0.371 | 0.472 | 0.364 | 0.463 | 0.365 | 0.471 | 0.367 |
| A1 | **0.515** | 0.375 | **0.473** | 0.368 | **0.474** | 0.360 | 0.470 | 0.388 | **0.470** | 0.360 | **0.485** | 0.376 |
| STG | **0.499** | 0.384 | **0.469** | 0.357 | **0.472** | 0.369 | **0.480** | 0.369 | 0.464 | 0.373 | **0.476** | 0.374 |
| MTG | 0.487 | 0.370 | 0.468 | 0.366 | 0.470 | 0.375 | **0.475** | 0.369 | **0.465** | 0.366 | 0.469 | 0.371 |
| ITG | 0.479 | 0.370 | 0.463 | 0.361 | 0.466 | 0.367 | 0.466 | 0.360 | 0.457 | 0.362 | 0.466 | 0.364 |
| Insula | 0.486 | 0.373 | 0.459 | 0.368 | 0.468 | 0.377 | **0.476** | 0.376 | 0.463 | 0.363 | **0.473** | 0.370 |
| TPJ | 0.474 | 0.345 | 0.468 | 0.349 | 0.466 | 0.361 | 0.463 | 0.347 | 0.462 | 0.347 | 0.467 | 0.350 |
| Temporal Pole | 0.484 | 0.384 | 0.467 | 0.362 | 0.455 | 0.373 | **0.475** | 0.365 | 0.459 | 0.362 | 0.469 | 0.351 |
| Sensorimotor | **0.491** | 0.373 | **0.470** | 0.363 | **0.471** | 0.375 | 0.470 | 0.362 | **0.466** | 0.366 | 0.472 | 0.370 |
| IFG | 0.480 | 0.370 | 0.465 | 0.365 | 0.465 | 0.366 | 0.466 | 0.360 | 0.462 | 0.359 | 0.467 | 0.364 |
| MFG | 0.475 | 0.367 | 0.457 | 0.374 | 0.462 | 0.364 | 0.466 | 0.355 | 0.453 | 0.373 | 0.462 | 0.376 |
| Hippocampus | 0.481 | 0.358 | 0.464 | 0.365 | 0.458 | 0.366 | 0.462 | 0.353 | 0.458 | 0.360 | 0.469 | 0.362 |
| Amygdala | 0.472 | 0.334 | 0.463 | 0.345 | 0.465 | 0.362 | 0.452 | 0.353 | 0.463 | 0.358 | 0.471 | 0.349 |

Table 4: Predictive alignment results for the **phrase** corpus. TThe first row reports the overall model–brain similarity $S(m,b)$; subsequent rows report the model–region similarity $S(m,b_r)$. Bold in the $S(m,b)$ row marks the highest value(s) across models/hemispheres; within each model, bold highlights the three highest $S(m,b_r)$ entries across all ROIs and hemispheres (ties bolded).

| | GPT-2 | | Gemma | | Gemma 2 | | Llama 2 | | Llama 3.1 | | GLM-4 | |
|---|---|---|---|---|---|---|---|---|---|---|---|---|
| | L | R | L | R | L | R | L | R | L | R | L | R |
| $S(m,b)$ | 0.454 | 0.356 | **0.465** | 0.362 | **0.465** | 0.366 | 0.446 | 0.350 | 0.451 | 0.354 | 0.449 | 0.349 |
| A1 | 0.448 | 0.351 | **0.478** | 0.343 | **0.482** | 0.345 | **0.451** | 0.365 | 0.452 | 0.333 | 0.443 | 0.344 |
| STG | **0.463** | 0.363 | **0.468** | 0.364 | **0.470** | 0.363 | 0.448 | 0.356 | **0.455** | 0.352 | **0.458** | 0.357 |
| MTG | 0.450 | 0.357 | 0.464 | 0.362 | 0.462 | 0.366 | 0.445 | 0.363 | 0.448 | 0.348 | 0.445 | 0.345 |
| ITG | 0.449 | 0.348 | **0.465** | 0.359 | 0.460 | 0.363 | 0.445 | 0.346 | 0.448 | 0.352 | 0.448 | 0.345 |
| Insula | 0.453 | 0.360 | 0.461 | 0.360 | 0.462 | 0.365 | 0.446 | 0.348 | 0.451 | 0.354 | **0.452** | 0.344 |
| TPJ | 0.454 | 0.350 | 0.462 | 0.361 | 0.460 | 0.361 | 0.444 | 0.343 | **0.450** | 0.350 | 0.448 | 0.349 |
| Temporal Pole | **0.461** | 0.371 | 0.464 | 0.363 | **0.464** | 0.365 | **0.450** | 0.347 | 0.448 | 0.354 | **0.452** | 0.344 |
| Sensorimotor | **0.461** | 0.371 | 0.464 | 0.363 | **0.464** | 0.365 | **0.450** | 0.347 | 0.448 | 0.354 | **0.452** | 0.344 |
| IFG | 0.450 | 0.351 | 0.454 | 0.362 | 0.453 | 0.358 | 0.437 | 0.343 | **0.454** | 0.353 | 0.448 | 0.347 |
| MFG | 0.451 | 0.328 | **0.466** | 0.333 | 0.460 | 0.361 | **0.455** | 0.330 | **0.455** | 0.353 | **0.457** | 0.351 |
| Hippocampus | 0.454 | 0.354 | 0.453 | 0.363 | 0.463 | 0.357 | 0.436 | 0.344 | **0.454** | 0.354 | 0.434 | 0.354 |
| Amygdala | 0.403 | 0.361 | 0.420 | 0.391 | 0.450 | 0.355 | 0.385 | 0.349 | 0.402 | 0.386 | 0.389 | 0.381 |

$F = 43.37$, $p < 0.001$, $\eta^2 = 0.008$; **phrase**: $F = 13.80$, $p < 0.001$, $\eta^2 = 0.008$), with consistently higher alignment in the left hemisphere. At the region level, FDR-controlled one-way ANOVAs showed significant between-model effects within left-dominant language cortices (e.g., STG, MTG, ITG) for both corpora. These findings align with the SRDM–RSA analyses, which likewise demonstrate significant between-model variation at both whole-brain and region-specific levels.

Mirroring the SRDM–RSA trend, the predictive results exhibit an almost identical trend for model upgrade comparison. In the **sentence** corpus, the left-hemisphere model–brain scores reproduce the RSA ordering—Gemma 2 exceeds Gemma ($S(m,b) = 0.469$ vs. $0.467$, L), while Llama 3.1 falls below Llama 2 ($0.463$ vs. $0.472$, L). In the **phrase** corpus, the gaps compress: Gemma 2 is essentially tied with Gemma (both $S(m,b) = 0.465$, L), and Llama 3.1 is slightly higher than Llama 2 ($0.451$ vs. $0.446$, L). This compression likely arises because phrase analyses use only phrase neurons and provide fewer, noisier frequency samples, which together reduce model separability. Taken together, the predictive encoding and SRDM–RSA analyses converge on a consistent, method-robust

conclusion: inter-model differences are statistically reliable at both the brain and region levels, and, at the brain level, model-upgrade trends are essentially identical across methods, with only minor fluctuations.

## C  Bilingual sentence- and phrase-level representations in LLMs

Previous studies have explored how LLMs handle different languages, concluding that while most neurons are shared across languages, a smaller subset of neurons is dedicated to processing specific languages [Tang et al., 2024]. But does this hold true for syntactic structure perception? This appendix provides insights into this question. It is important to note that the GPT-2 model used in this study was pre-trained on a Chinese corpus using the Universal Encoder Representations (UER) framework [Zhao et al., 2019], equipping it with the capability to process both Chinese and English text effectively. This stands in contrast to the original GPT-2 model [Radford et al., 2019], which lacks the ability to handle Chinese text.

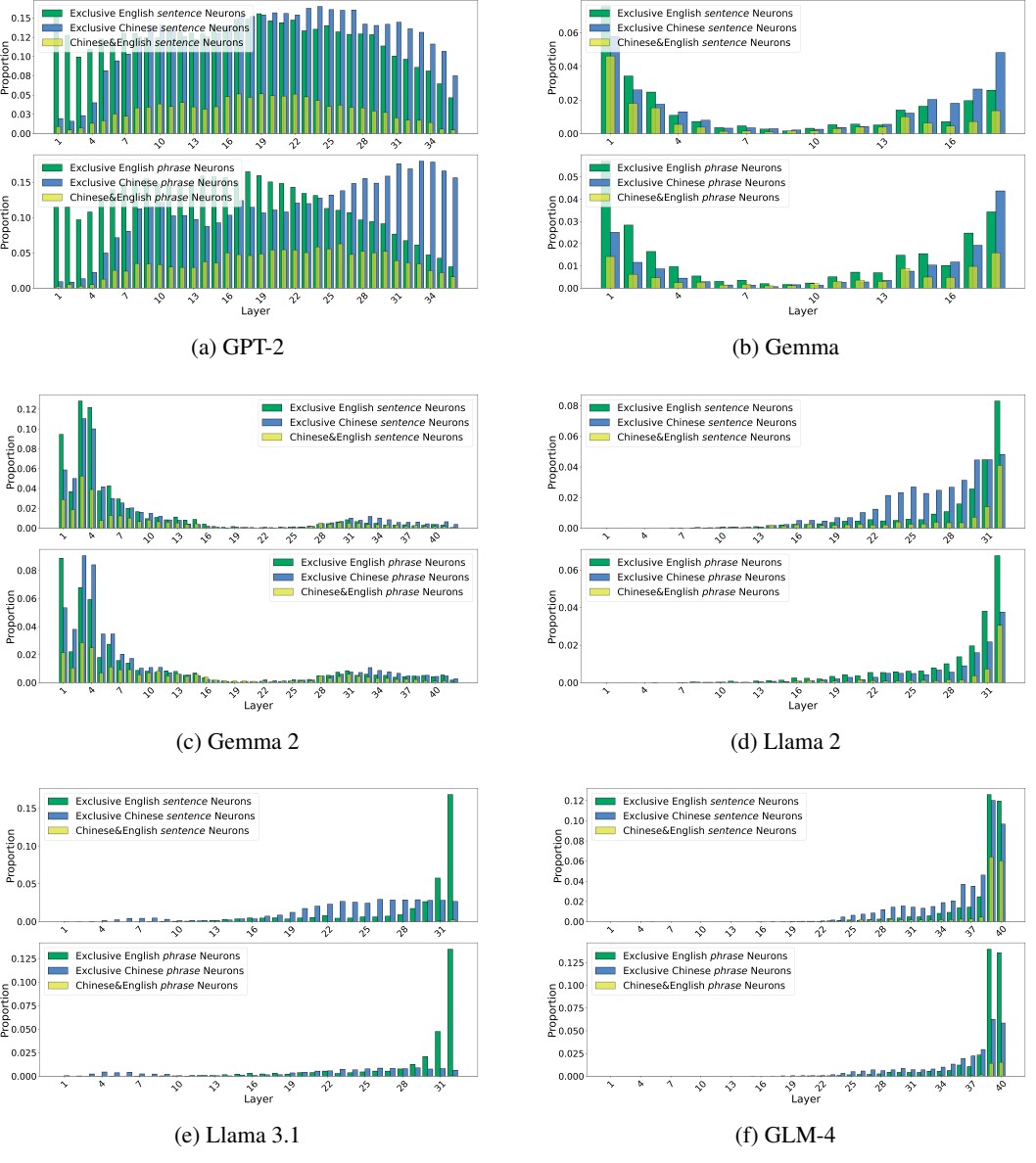

Figure 7: Cross-language neural representations extracted from five multilingual models (Gemma, Gemma 2, Llama 2, Llama 3.1, and GLM-4) depicting syntactic processing capabilities.

We used semantically matched Chinese and English syntactic corpora in this bilingual experiment (see Tables 6 and 7). The results in Figure 7 suggest that language-specific syntactic MLP neurons (i.e., exclusive sentence/phrase neurons) tend to cluster toward the final layers of Llama 2, Llama 3.1, and GLM-4, with the proportion of bilingual neurons (Chinese & English) increasing progressively in deeper layers. In contrast, Gemma and Gemma 2 display different patterns. In Gemma, both language-specific and bilingual neurons are found not only in the deeper layers but also in the initial layers, whereas this is only observed in the early layers of Gemma 2. GPT-2, on the other hand, exhibits a more balanced distribution, with both language-specific and bilingual neurons present in almost every layer in roughly equal proportions. Furthermore, English-specific neurons are more prominent in the early layers, while Chinese-specific neurons are more concentrated in the later layers. Interestingly, Llama 3.1 shows a notably lower count of Chinese-specific neurons compared to English-specific neurons in the final layer, and fewer Chinese & English neurons than the other five LLMs. Although Llama 3.1 was pre-trained on 176 languages [Dubey et al., 2024], it appears to have less specialization in Chinese, which may explain the reduced presence of Chinese-specific neurons and, consequently, fewer bilingual neurons.

## D    Contribution Ratios of LLMs

To further investigate the role of specific brain ROIs in syntactic processing, we introduced the contribution ratio $CR_r$. The contribution ratio highlights which brain ROIs contribute most significantly to the syntactic alignment between LLMs and the human brain. Fixing a model layer, this metric quantifies the influence of each brain ROIs by calculating the proportion of channels from a given region within the top 100 most relevant channels, normalized by the overall representation of the ROIs. The contribution ratio is defined as:

$$CR_r(L_j) = \frac{N_r^{\text{top}}(L_j)/N^{\text{top}}}{N_r^{\text{total}}/N^{\text{total}}}, \tag{8}$$

where $N_r^{\text{top}}(L_j)$ is the number of channels in region $r$ within the top 100 channels in terms of the LLM layer $L_j$, $N^{\text{top}}$ is the total number of top channels, which is specified as 100 in this case, $N_r^{\text{total}}$ is the total number of channels in region $r$, and $N^{\text{total}}$ is the total number of brain channels.

We present the contribution ratio results for six LLMs used in this study: GPT-2, Gemma, Gemma 2, Llama 2, Llama 3.1, and GLM-4. Specifically, the contribution ratio for each model was calculated based on the number of top 100 significant channels within each brain ROIs, as described in Appendix A. Below, we present the results for both the left (L) and right (R) hemispheres of each model (See Figures 15, 16, 17, 18, 19 and 20). These figures offer further insights into how different LLMs align with human brain ROIs in terms of syntactic processing.

From these figures, we observe that across all LLMs, regions such as A1 and STG in the left hemisphere, and the Insula, Temporal Pole, and Amygdala in the right hemisphere contribute more significantly to the alignment with human brain syntactic processing. These regions are known to be involved in language-specific processes in the human brain, particularly in the left hemisphere, where the STG and A1 are crucial for auditory and syntactic processing. Overall, these contribution profiles suggest that these models may be capturing aspects of hierarchical syntactic structures in ways that are functionally similar to human neural mechanisms. The Insula, Temporal Pole, and Amygdala, though not traditionally highlighted as primary language regions, may also play supporting roles in language comprehension, possibly through emotion and memory-related pathways.

## E    HFTP on naturalistic corpus

Beyond validating HFTP with tightly controlled four-syllable/word sequences, we show that it generalises to naturalistic text spanning everyday dialogue, news reports, literature, and poetry. We tested eight- and nine-syllable Chinese corpora alongside matched English 8- and 9-word corpora (see Tables 8–9, 10–11). Spectral analysis reveals four pronounced peaks in each condition. For the eight-syllable set, peaks arise at 0.5, 1.0, 1.5 and 2.0 Hz, corresponding to the full–sentence envelope, the canonical 4-character phrase rhythm, an intermediate 2–3-beat grouping around 1.5 Hz, and the ubiquitous 2-character lexical rhythm (Figure 8), with the English 8-word corpus exhibiting the

same four-peak pattern at the corresponding sentence and phrase rates (Figure 10). In nine-syllable sentences the peaks shift to $\sim 0.44, 0.89, 1.33$, and $1.78$ Hz: the lowest peak reflects the whole sentence, $1.33$ Hz aligns with abundant 3-character phrases, $1.78$ Hz captures rapid two-to-three-character alternations, and the $0.89$ Hz component indexes a prosodic half-sentence "breath group" of four–to–five characters (Figure 9), and the English 9-word corpus shows the analogous four peaks near these frequencies (Figure 11).

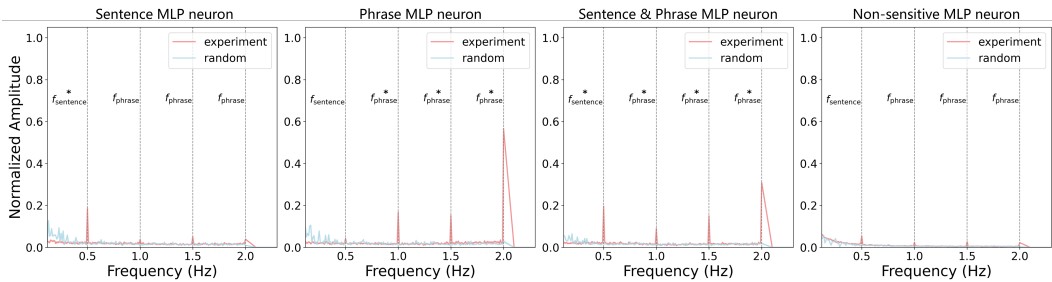

Figure 8: Hierarchical frequency patterns of MLP neurons, using a naturalistic Chinese 8-syllable corpus, selectively represent sentence features, phrase features, shared features of both, and non-sensitive features (from left to right).

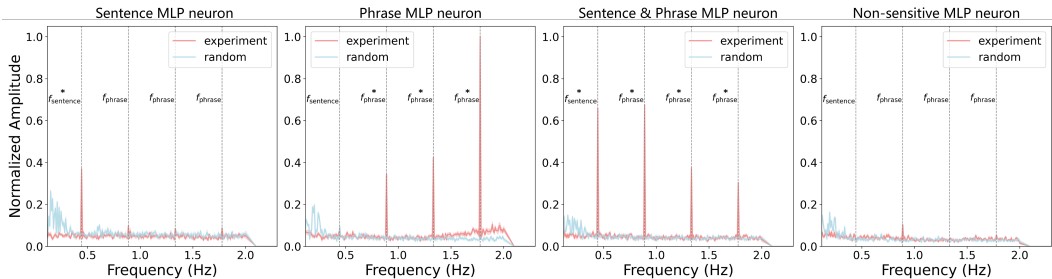

Figure 9: Hierarchical frequency patterns of MLP neurons, using a naturalistic Chinese 9-syllable corpus, selectively represent sentence features, phrase features, shared features of both, and non-sensitive features (from left to right).

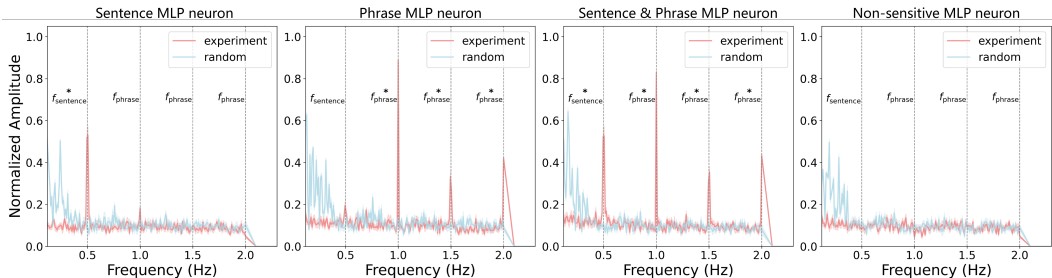

Figure 10: Hierarchical frequency patterns of MLP neurons, using a naturalistic English 8-word corpus, selectively represent sentence features, phrase features, shared features of both, and non-sensitive features (from left to right).

We further evaluated HFTP on Chinese and English Wikipedia 8-syllable/word corpora (Figures 12–13). In both languages a clear sentence peak appears at $0.5$ Hz, and phrase-rate peaks persist at $1.0, 1.5$, and $2.0$ Hz; however, relative to non-Wikipedia corpora the peaks are attenuated, and the separation between experiment and random controls is smaller. Although sequence length was strictly controlled to eight syllables/words, Wikipedia text is less regular in content: mixed scripts/orthographies (e.g., simplified vs. traditional in Chinese), frequent abbreviations and alphanumeric tokens, bibliographic fragments and formulaic titles, heterogeneous named entities,

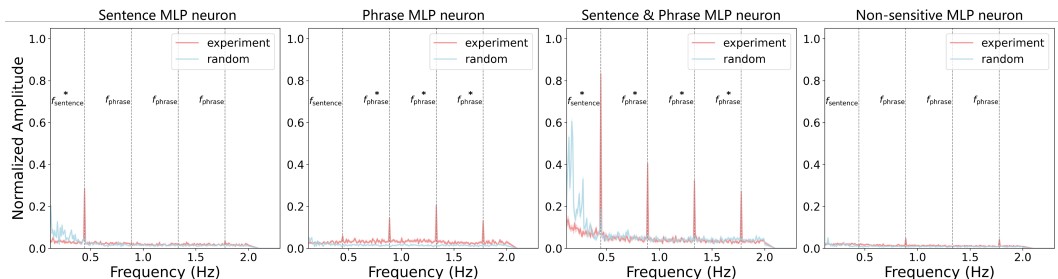

Figure 11: Hierarchical frequency patterns of MLP neurons, using a naturalistic English 9-word corpus, selectively represent sentence features, phrase features, shared features of both, and non-sensitive features (from left to right).

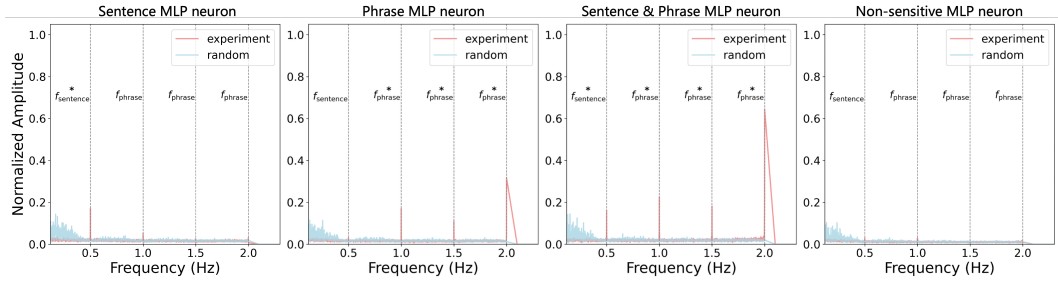

Figure 12: Hierarchical frequency patterns of MLP neurons, using a Chinese Wikipedia 8-syllable corpus, selectively represent sentence features, phrase features, shared features of both, and non-sensitive features (from left to right).

inconsistent prosodic phrasing, etc. These factors reduce cross-sentence periodicity and weaken harmonic reinforcement, thereby weakening tracking at sentence- and phrase-rate rhythms.

We speculate that the four dominant frequencies arise from tokenization statistics and prosodic templating shared across languages: frequent two- and three-token words and binomial/trinomial chunks in English, and abundant 2-/3-character words and 4-character idioms in Chinese; both yield stable 2- and 3-beat groupings, while clause-level phrasing produces a half-sentence rhythm. Transformer layers further reinforce harmonics of the basic word cycle. For example, in the nine-syllable corpus, a 3-beat unit around $1.33$ Hz naturally gives rise to a higher harmonic near $1.78$ Hz. The half-sentence peak at $0.89$ Hz emerges because speakers often place a prosodic break near the midpoint of nine-character or nine-word clauses, creating a stable sub-sentence rhythm that the model entrains to. Because the underlying rhythmic structure is shared, we believe HFTP is highly likely to generalize to other character-centric languages (Japanese, Korean) and to space-delimited alphabetic languages (French, German).

## F   Preliminary bilingual HFTP test on native Chinese speakers

As noted above, our HFTP analysis was originally conducted on native Chinese speakers listening to a Chinese corpus. To probe cross-linguistic generalization in human sEEG, we constructed bilingual materials by taking the Chinese corpus from [Sheng et al., 2019] and manually translating it into four-word English sequences. All experimental settings (e.g., presentation duration, sampling rate) were identical to those in the original sEEG experiment. We report preliminary results from one native Chinese speaker who listened to both the Chinese and the English corpora. Hierarchical frequency patterns for the Chinese corpus closely match those in Figure 5, and the corresponding English results are shown in Figure 14.

In the English condition, we observe the canonical hierarchical fingerprint: sentence-selective channels peak at $1$ Hz, phrase-selective channels peak at $2$ Hz, channels sensitive to both show peaks at both frequencies, and non-sensitive channels show no systematic peaks. This replication in English indicates that the frequency-tagged response generalizes across languages when the hierarchical

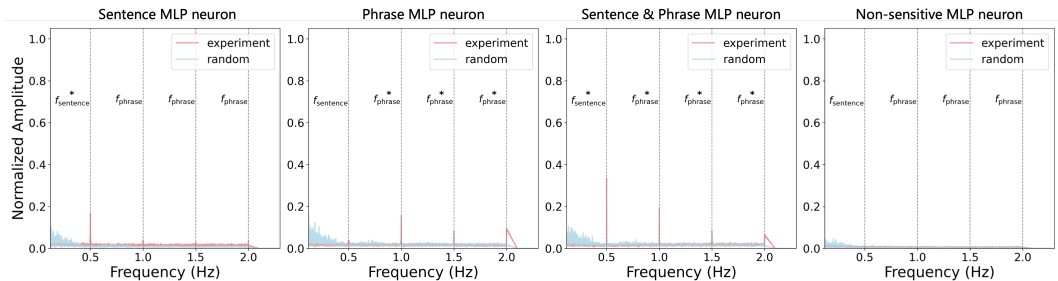

Figure 13: Hierarchical frequency patterns of MLP neurons, using a English Wikipedia 8-word corpus, selectively represent sentence features, phrase features, shared features of both, and non-sensitive features (from left to right).

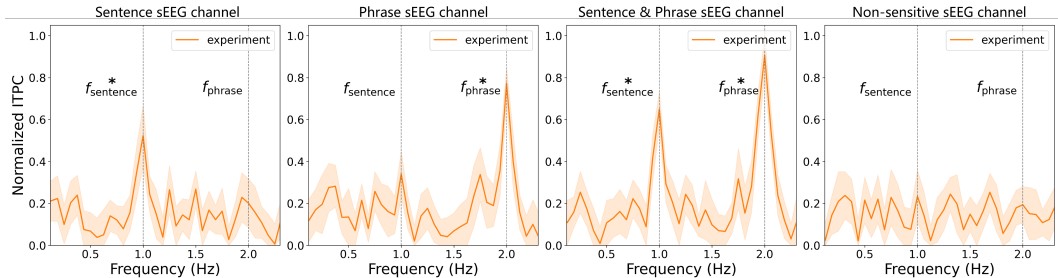

Figure 14: Hierarchical frequency patterns of sEEG channels from one participant, using four-word English sequence, selectively represent sentence features, phrase features, shared features of both and non-sensitive feature (from left to the right). Shaded bands show $\pm 1$ s.e.m. computed within each channel by bootstrapping across sliding-window ITPC estimates.

structure of the input is preserved, arguing that HFTP tracks syntactic organization rather than language-specific acoustics or phonotactics. While promising, these observations are based on one participant and should be treated as proof-of-principle rather than definitive. Future work will expand the sample, balance materials across languages, and use more robust statistical analyses to quantify cross-linguistic effect sizes.

# G   Model details

In this appendix, we present the details of the LLMs used in this study. Table 5 summarizes key parameters, including model size, number of layers, attention heads, and MLP neurons. Notably, the GPT-2 variant used here was pre-trained on a Chinese corpus, enabling it to process both Chinese and English; by contrast, the original GPT-2 does not support Chinese.

Table 5: Comparison of model parameters.

| Model | Size | Layers | Attention heads | MLP neurons |
|---|---|---|---|---|
| GPT-2 [Radford et al., 2019] | 774M | 36 | 20 | 5120 |
| Gemma [Team et al., 2024a] | 7B | 28 | 16 | 24576 |
| Gemma 2 [Team et al., 2024b] | 9B | 42 | 16 | 14336 |
| Llama 2 [Touvron et al., 2023] | 7B | 32 | 32 | 11008 |
| Llama 3.1 [Dubey et al., 2024] | 8B | 32 | 32 | 14336 |
| GLM-4 [GLM et al., 2024] | 9B | 40 | 32 | 13696 |

# H Syntactic corpora

For LLM experiments, we used a diverse suite of corpora spanning languages, structure, and length. Specifically, we used Chinese four-syllable and English four-word syntactic corpora adapted from [Ding et al., 2016] (Tables 6, 7), alongside human-generated naturalistic Chinese corpora (eight- and nine-syllable; Tables 8, 9) and parallel English counterparts (eight- and nine-word; Tables 10, 11). These corpora were used to assess model sensitivity to hierarchical structure. In addition, we constructed Wikipedia-derived Chinese and English 8-syllable/word corpora for out-of-domain validation; these corpora are available on `https://github.com/LilTiger/HFTP`.

Table 6: Chinese syntactic corpus.

| Four-syllable sequences | | | | |
|---|---|---|---|---|
| 老牛耕地 | 朋友请客 | 厨师做饭 | 竹鼠啃笋 | 农民种菜 |
| 青草发芽 | 和尚念经 | 老师讲课 | 鲸鱼喷水 | 绵羊吃草 |
| 英雄救火 | 游客爬山 | 鸭子划水 | 蜘蛛结网 | 祖父下棋 |
| 医生看病 | 护士打针 | 母鸡下蛋 | 行人过街 | 法官判案 |
| 狮子吃肉 | 老鹰捕鱼 | 蜜蜂采花 | 小孩读书 | 司机开车 |
| 画家作画 | 船夫摇桨 | 诗人吟诗 | 麻雀筑巢 | 猴子摘桃 |
| 渔夫撒网 | 骆驼饮水 | 狐狸捕鼠 | 海豹顶球 | 小猫抓鱼 |
| 老马拉车 | 鸽子衔枝 | 孩童拾贝 | 雏鸡啄米 | 山雀捉虫 |
| 青鸟啄木 | 樵夫砍柴 | 黑熊爬树 | 土狼挖洞 | 军鸽传信 |
| 燕雀喂仔 | 野猪拱地 | 渔民划船 | 蚯蚓钻土 | 蚕蛾吐丝 |

Table 7: English syntactic corpus.

| Four-word sequences | | |
|---|---|---|
| Old ox plows field | A friend invites guests | The chef cooks dinner |
| Bamboo rat gnaws shoots | A farmer plants vegetables | Green grass sprouts up |
| The monk chants scriptures | A teacher gives lecture | The whale spouts water |
| The sheep eats grass | A hero extinguishes fire | The tourist climbs mountain |
| Pond duck paddles water | A spider spins web | My grandfather plays chess |
| The doctor treats patients | The nurse administers vaccine | A hen lays eggs |
| The pedestrian crosses street | The judge decides case | A lion eats meat |
| An eagle catches fish | Honey bee gathers nectar | The child reads books |
| The driver operates vehicle | A painter creates art | The boatman rows oars |
| The poet recites verses | The sparrow builds nest | A monkey picks peaches |
| A fisherman casts net | The camel drinks water | The fox catches mice |
| A seal balances ball | A kitten catches fish | Old horse pulls cart |
| A dove carries branch | The child collects shells | A chick pecks grain |
| A titmouse catches insects | The bluebird pecks wood | A woodcutter chops firewood |
| Black bear climbs tree | A coyote digs burrow | War pigeon delivers messages |
| The swallow feeds chicks | Wild boar roots earth | A fisherman rows boat |
| The earthworm burrows soil | Silkworm moth spins silk | |

For the human brain experiment, we utilized two Chinese corpora: the **sentence** and **phrase** corpora. To ensure consistent analysis of syntactic processing across both LLMs and the human brain, the same corpora were applied to the alignment experiment. These corpora originated from [Sheng et al., 2019]. Participants received a brief oral instruction to listen carefully to each stimulus, completed 40 trials per condition (sentence, phrase, and random), and were compensated approximately ¥200 for full participation, with pay prorated if they completed only a subset of trials.

# I Brain ROIs

As discussed in Section 3.3, we reorganized the original sEEG data by grouping the Automated Anatomical Labeling (AAL) annotations into newly defined brain ROIs for our experi-

Table 8: Naturalistic Chinese corpus.

| Eight-syllable sequences | | |
|---|---|---|
| 森林火势得到控制。 | 列车准点抵达站台。 | 今晚，一起看电影吗？ |
| 中央推出税收优惠。 | 请稍等，我来核对价。 | 空调滤网需要更换。 |
| 研究论文成功发表。 | 画展现场气氛静谧。 | 港口货轮密集靠泊。 |
| 明天，早饭想吃啥呢？ | 请问洗手间在哪呀？ | 他缓缓走入雨巷中。 |
| 古桥石栏苔痕深留。 | 图书馆今天人较多。 | 教育部发布新课程。 |
| 快看，雨停了出门吧！ | 市场需求逐步回暖。 | 点单吗？我们有套餐。 |
| 快递包裹正在派送。 | 图书销量榜单更新。 | 雨夜街灯映成河影。 |
| 证券监管再次收紧。 | 服务员，来两杯绿茶！ | 警方破获网络诈骗。 |
| 同学，你借我下笔吧？ | 科研团队揭量子谜。 | 春雷滚过江南田畔。 |
| 旅行社推出特价游。 | 学生提交毕业论文。 | 他戴上耳机工作中。 |
| 老师，这题怎么写呢？ | 他整理旧照片回忆。 | 海浪轻拍沙滩细岸。 |
| 亲爱的，晚餐想吃啥？ | 喂，你现在到哪里了？ | 股市午盘震荡收高。 |
| 你好，咖啡需要糖吗？ | 桂花香飘整条街上。 | 日出染红东海天际。 |
| 月光洒落在城墙上。 | 乡村集市热闹开张。 | 记者现场连线报道。 |
| 航班因雾全部延误。 | 数据中心全面升级。 | 雨天，道路易积水患。 |
| 智能巴士全线运营。 | 剧院上演经典芭蕾。 | 博物馆新增夜场票。 |
| 南部遭遇强降雨灾。 | 想不到你如此嚣张。 | |

Table 9: Naturalistic Chinese corpus.

| Nine-syllable sequences | | |
|---|---|---|
| 临床试验数据公布了。 | 姐姐，这裙子有蓝色吗？ | 软件更新，漏洞已修复。 |
| 晚上一起吃寿司如何？ | 孩子，慢点吃别噎到啊。 | 付款可用微信，是不是？ |
| 医护人员彻夜守病房。 | 智能家居系统升级中。 | 校园社团招募新成员。 |
| 我们有可乐、果汁、豆浆。 | 高中将设人工智能课。 | 你好，考试时间改了吗？ |
| 摄影展聚焦城市微光。 | 他翻身查看夜空星图。 | 街角花店玫瑰已售罄。 |
| 师傅，去火车站多少钱？ | 大雨突临，烟花秀取消。 | 他低声读完那封旧信。 |
| 科研数据平台上线啦。 | 社区篮球赛今晚开哨。 | 市场监管局突击检查。 |
| 木星再添两颗小卫星。 | 疏影横斜暗上书窗敲。 | 晚餐菜单更新完毕了。 |
| 明早七点机场见，好吗？ | 老板，这条鱼再便宜点？ | 卫星成功捕捉极光影。 |
| 雨滴敲击玻璃声清脆。 | 电商推广使用绿色包。 | 他轻扣门板等待回应。 |
| 喂，你到公司门口了吗？ | 旅行箱在传送带循环。 | 服务员，账单麻烦拿来。 |
| 凌晨街头灯火渐稀少。 | 国家队再夺巴黎首金。 | 音乐渐缓舞步更轻盈。 |
| 早高峰地铁挤满乘客。 | 新剧首播口碑节节攀。 | 灯光映照湿润青石路。 |
| 医生，这药饭前还是后？ | 请坐，这里视线最好哦。 | 你好，请填写到访登记。 |
| 咖啡豆飘散焦糖香气。 | 请稍候，我去取电影票。 | 他轻敲键盘修改代码。 |
| 图书销量排行榜刷新。 | 同学，笔记本借我一下？ | 志愿者分发食物物资。 |
| 快递无人签收被退回。 | 博主分享无人机航拍。 | |

ments [Rolls et al., 2015]. In this appendix, we provide the full names of AAL regions, the corresponding AAL labels used in the sEEG data, and their mapped brain ROIs in Table 12.

Table 10: Naturalistic English corpus.

| Eight-word sequences | |
| --- | --- |
| With malice toward none, with charity toward all. | Tonight, shall we watch the meteor shower together? |
| A man can be destroyed but not defeated. | Excuse me, where's the nearest restroom around here? |
| The library felt crowded during rainy examination week. | Market demand seems gradually rebounding during this quarter. |
| Book sales rankings were refreshed early this morning. | Waiter, two cups of hot green tea please. |
| Quantum research team unveiled perplexing entanglement results yesterday. | Students submitted their final graduate theses by noon. |
| He sorted aging photographs, reminiscing about bygone days. | Hello, where are you right now, my friend? |
| Moonlight spilled gently across the ancient city walls. | Flights were delayed nationwide due to dense fog. |
| Autonomous buses now operate on every urban route. | Southern region suffered severe flooding after relentless rain. |
| Plum blossoms scented the courtyard with delicate sweetness. | Please hold on, I'll check your booking details. |
| Central bank announced fresh stimulus to boost economy. | Port cranes unloaded containers under brilliant afternoon sky. |
| He slowly wandered into the rainy narrow alleyway. | Education ministry announced updated national curriculum guidelines today. |
| Ready to order? We have today's special combo. | Rainy night streetlights shimmered like a glowing river. |
| Police dismantled an extensive online fraud network operation. | Spring thunder rolled over rice paddies in Jiangnan. |
| He put on headphones and focused on coding. | Gentle waves lapped softly against the sandy shoreline. |
| Could you use some sugar in your coffee? | The rural marketplace opened lively at dawn today. |
| The data center completed a comprehensive system upgrade. | The theater staged a timeless classical ballet tonight. |
| I never imagined you could be this arrogant. | City skyline glittered beneath a crisp winter moon. |
| Train whistle echoed across fields drenched with mist. | Research findings published in a prestigious journal today. |
| Tomorrow, what would you like for breakfast, friend? | Ancient stone bridge retained mossy grooves through centuries. |
| Look, the storm passed; let's explore the streets. | Parcel delivery is currently out for neighborhood distribution. |
| Financial regulators tightened oversight on speculative securities trading. | Hey classmate, may I borrow your pen briefly? |
| Travel agency launched discounted spring break tour packages. | Teacher, could you explain this problem once more? |
| Darling, what would you like for dinner tonight? | Osmanthus fragrance drifted along the entire narrow street. |
| Reporter delivered live coverage from bustling downtown square. | On rainy days, roads easily accumulate dangerous puddles. |
| Museum introduced extended evening hours for visitors' convenience. | Night market vendors grilled skewers over glowing charcoal. |

## Table 11: Naturalistic English corpus.

| Nine-word sequences | |
| --- | --- |
| Clinical trial data were publicly released this morning nationwide. | Shall we share sushi together tonight by the river? |
| War is peace, freedom is slavery, ignorance is strength. | We offer cola, soda, juice, and soy milk today. |
| Photography exhibition focuses on city's hidden pockets of light. | Driver, how much is the fare to the station? |
| Scientific data platform finally went live to the public. | Jupiter just gained two additional tiny moons last week. |
| See you at the airport tomorrow seven sharp, alright? | Raindrops drummed against windowpanes in a crisp cadence tonight. |
| Hey, have you reached the company entrance yet today? | Early morning city lights gradually faded as traffic intensified. |
| Morning rush subway overflowed with restless hurried commuters again. | Doctor, should this new medicine be taken before dinner? |
| Coffee beans release subtle caramel aroma throughout the cafe. | Book sales leaderboard refreshed, surprising many independent authors today. |
| Undelivered parcels were returned after nobody signed for them. | Sister, does this blue dress come in medium size? |
| Child, slow down and chew, avoid choking on food. | Smart home automation system is currently installing new firmware. |
| High school will introduce artificial intelligence elective next year. | He rolled over, consulting a star chart at midnight. |
| Sudden downpour forced cancellation of tonight's fireworks display completely. | Community basketball tournament tips off under bright evening floodlights. |
| Slanting shadows tapped an old window softly at dusk. | Boss, could this fish be a little cheaper please? |
| Ecommerce campaign now promotes sustainable green packaging materials nationwide. | Luggage carousel kept circling the unattended silver suitcase indefinitely. |
| National team claimed its first gold medal in Paris. | New drama premiere received soaring reviews across social media. |
| Please sit here; the view remains the best available. | Please wait, I will retrieve our cinema tickets now. |
| Classmate, may I borrow your notebook for a moment? | Influencer shared breathtaking drone footage of mountain sunrise yesterday. |
| Software update completed, critical vulnerabilities have been fixed already. | Payment through WeChat is available, is that alright sir? |
| Campus club seeks enthusiastic freshmen to join this semester. | Hello, has the final exam schedule been changed recently? |
| Corner flower shop sold out all roses before noon. | He whispered while finishing that fading wartime love letter. |
| Market supervision bureau conducted an unexpected compliance inspection today. | Dinner menu has been fully updated for the evening. |
| Satellite captured brilliant aurora images above polar orbit yesterday. | He gently knocked, waiting patiently for someone to respond. |
| Please bring the bill, waiter, we are finished here. | Music slowed, dancers embraced lighter steps beneath dimmed lights. |
| Lantern glow reflected onto slick cobblestones after evening drizzle. | Hello, kindly complete the visitor registration form at reception. |
| He tapped the keyboard softly, refining his source code. | Volunteers distributed food supplies to displaced families after flood. |

Table 12: Automated Anatomical Labeling (AAL) annotations from the original sEEG data, along with their mapped brain ROIs. Note that the regions are distinguished by the left and right hemispheres.

| AAL Full Name | AAL Label | ROI |
|---|---|---|
| Heschl Gyrus | Heschl | A1 |
| Superior Temporal Gyrus | Temporal_Sup | STG |
| Middle Temporal Gyrus | Temporal_Mid | MTG |
| Inferior Temporal Gyrus | Temporal_Inf | ITG |
| Parahippocampal Gyrus | ParaHippocampal | ITG |
| Fusiform Gyrus | Fusiform | ITG |
| Insular Cortex | Insula | Insula |
| Angular Gyrus | Angular | TPJ |
| Supramarginal Gyrus | SupraMarginal | TPJ |
| Inferior Parietal Lobule | Parietal_Inf | TPJ |
| Superior Temporal Pole | Temporal_Pole_Sup | Temporal_Pole |
| Middle Temporal Pole | Temporal_Pole_Mid | Temporal_Pole |
| Paracentral Lobule | Paracentral_Lobule | Sensorimotor |
| Supplementary Motor Area | Supp_Motor_Area | Sensorimotor |
| Rolandic Operculum | Rolandic_Oper | Sensorimotor |
| Precentral Gyrus | Precentral | Sensorimotor |
| Postcentral Gyrus | Postcentral | Sensorimotor |
| Inferior Frontal Gyrus, Opercular part | Frontal_Inf_Oper | IFG |
| Inferior Frontal Gyrus, Triangular part | Frontal_Inf_Tri | IFG |
| Inferior Frontal Gyrus, Orbital part | Frontal_Inf_Orb | IFG |
| Middle Frontal Gyrus | Frontal_Mid | MFG |
| Middle Frontal Gyrus, Orbital part | Frontal_Mid_Orb | MFG |
| Hippocampus | Hippocampus | Hippocampus |
| Amygdala | Amygdala | Amygdala |

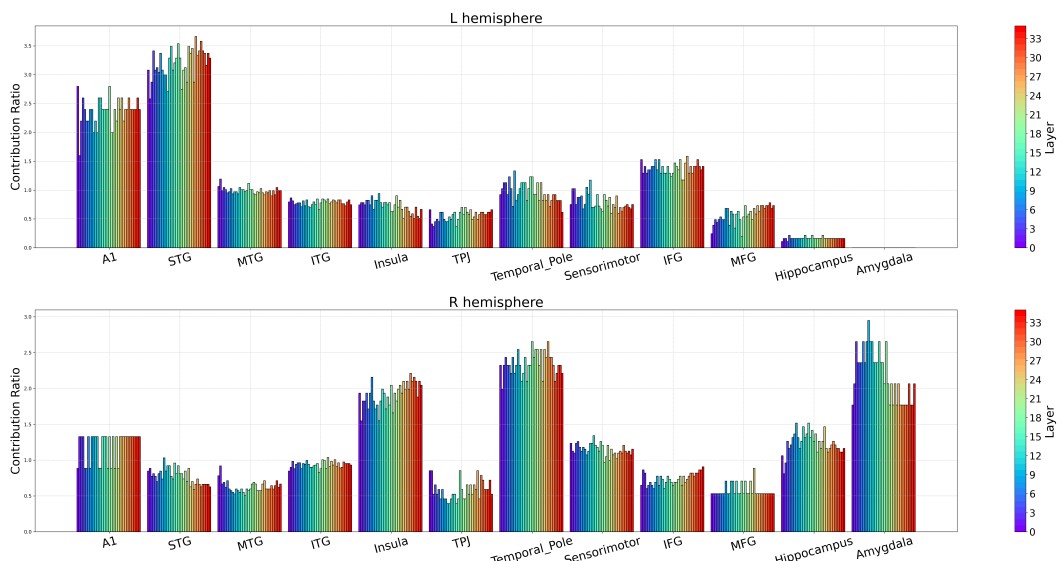

Figure 15: Contribution ratios for GPT-2: Left hemisphere (top) and Right hemisphere (bottom).

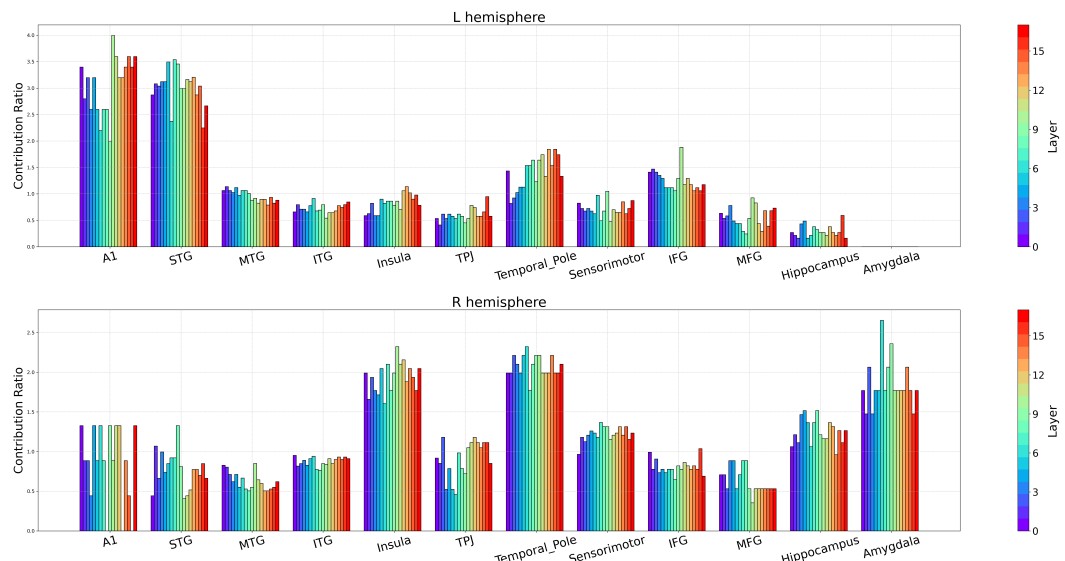

Figure 16: Contribution ratios for Gemma: Left hemisphere (top) and Right hemisphere (bottom).

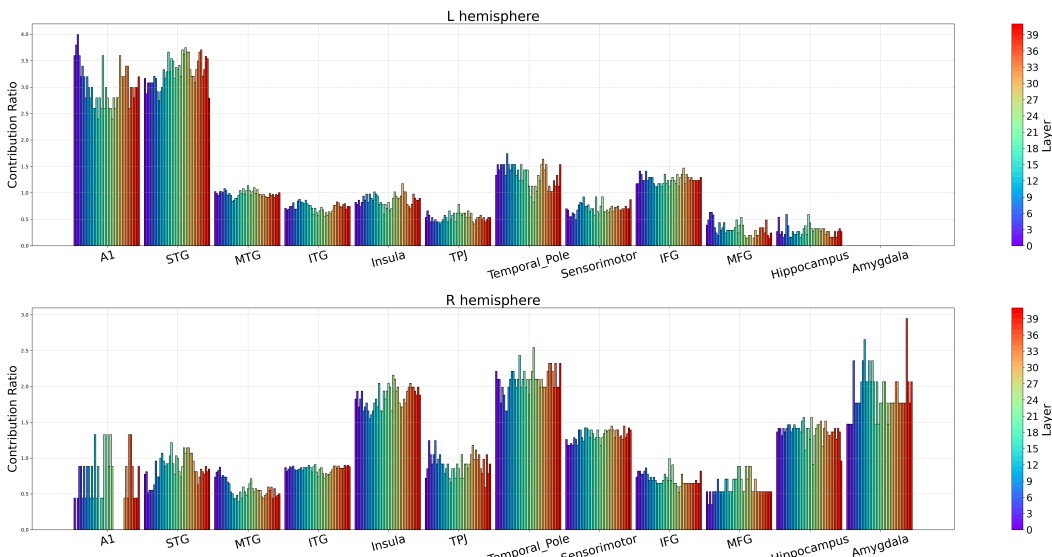

Figure 17: Contribution ratios for Gemma 2: Left hemisphere (top) and Right hemisphere (bottom).

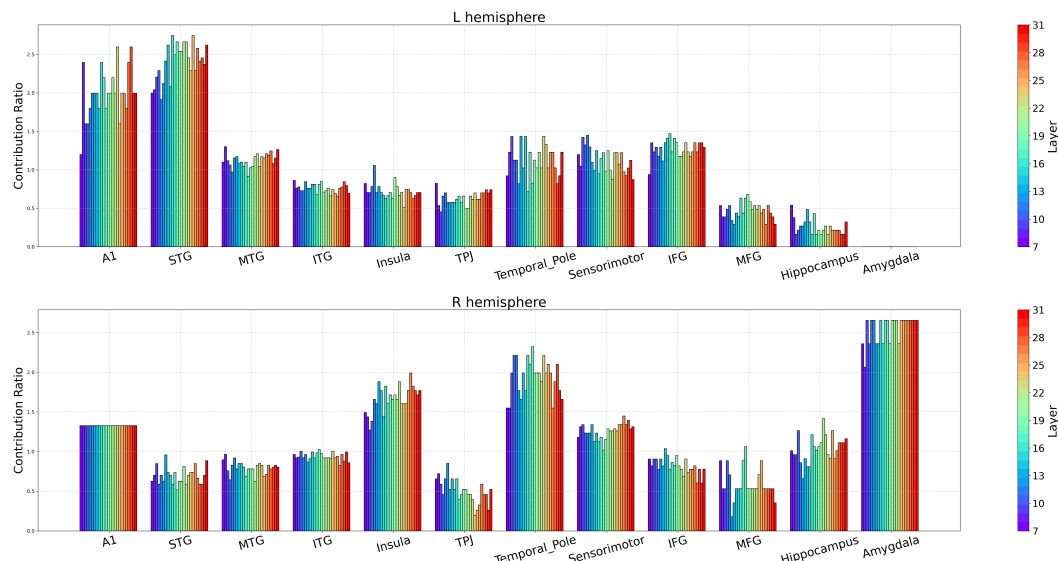

Figure 18: Contribution ratios for Llama 2: Left hemisphere (top) and Right hemisphere (bottom).

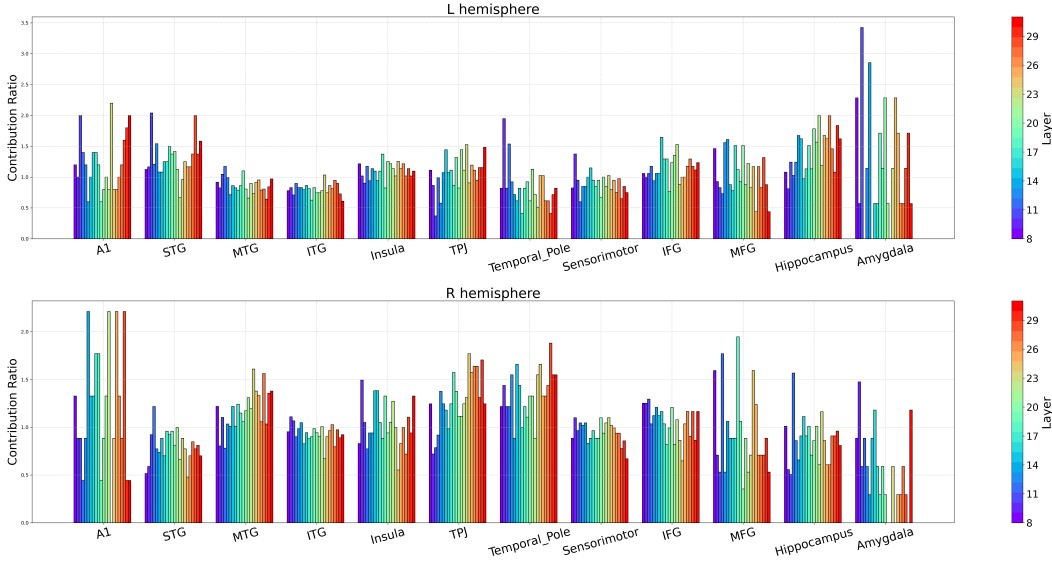

Figure 19: Contribution ratios for Llama 3.1: Left hemisphere (top) and Right hemisphere (bottom).

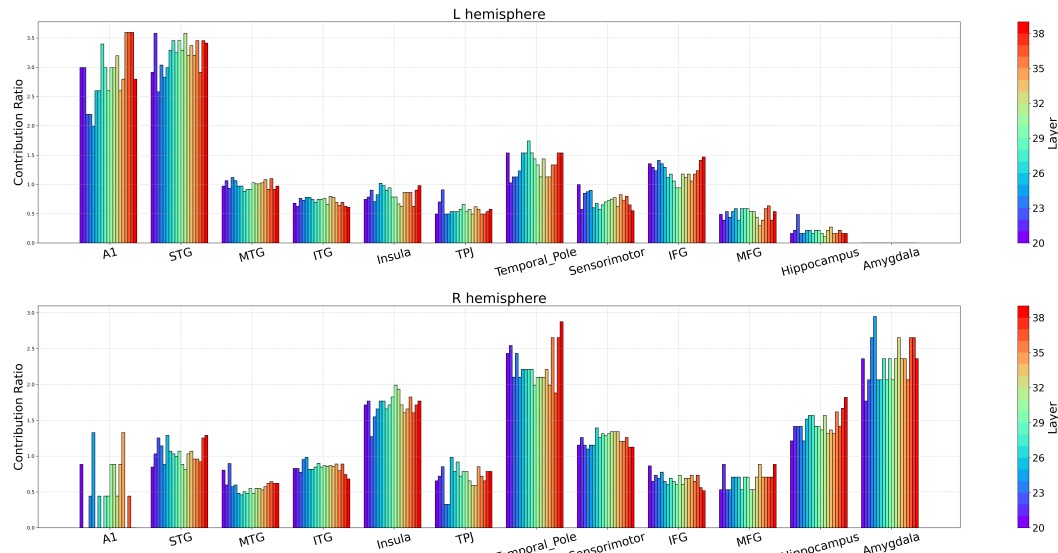

Figure 20: Contribution ratios for GLM-4: Left hemisphere (top) and Right hemisphere (bottom).

