# OpenReview forum: "Hierarchical Frequency Tagging Probe (HFTP): A Unified Approach to Investigate Syntactic Structure Representations in Large Language Models and the Human Brain"
_NeurIPS.cc/2025/Conference — NeurIPS 2025 poster_

### Official Review · Reviewer_2nNc · 2025-07-01

**Clarity:** 2
**Significance:** 2
**Originality:** 3
**Rating:** 4
**Confidence:** 3

**Summary:**

The authors present an extension of the experimental method of Ding et al. (2016) to both a  dataset of human subjects with intracranial sEEG recording and a fleet of text-based Transformer language models. This method revealed, in both Transformer models and brains, individual units (or electrodes) which exhibited structure tracking (=frequency tracking) at the phrase- and sentence-level. They found that different language models exhibited vastly different distributions of phrase- and sentence-selectivity, but that within any model, the distribution of phrase- and sentence-level neurons is strongly correlated across layers. This latter fact does not hold in a similar comparison applied across brain regions, suggesting a distinct gross architecture for syntactic processing between models and the human brain.

**Questions:**

Please minimally address the points given with asterisks above.

**Ethical Concerns:**

["NO or VERY MINOR ethics concerns only"]

**Final Justification:**

The authors have shared an impressive amount of extra work in their rebuttal. The structural probe-style analysis along with statistical controls settle some of my questions, and I also appreciate the naturalistic English experiment presented in response to other reviewers. Answers to methodological questions are also satisfactory. I will raise my score from 2 to 4.

**Limitations:**

Yes.

**Quality:**

2

**Strengths And Weaknesses:**

This is an ambitious paper which transfers a popular method from cognitive neuroscience to AI, and draws broad conclusions about an architectural dissociation between models and humans as a result. The experiments are extensive on the modeling side, covering a large class of LLMs.

However, there are some major analytic and experimental issues that make this paper difficult to accept at this point. These are given below with asterisks. Other conceptual / presentational issues are also highlighted.

- 4.1 MLP neurons represent sentences and phrases
	- You find a strong correlation between the number of sentence and phrase neurons within-model across layers (L254–257). This is interesting. How do you reconcile this with existing claims that LLMs distribute different syntactic functions across distinct layers (e.g. Tenney et al. (2019))?
	- \* L237–246: The interpretation of differences between phrase- and sentence-selectivity layer distributions across models is pretty unsatisfying ("reflecting a more integrated approach ... suggesting a balanced intermediate strategy ... indicating an emphasis on initial-stage syntactic processing"). These labels aren't very meaningful without further analysis. Given that this is a core finding of your paper (and part of the critical mismatch with the brain findings) I think you need to do more work to detail what kind of process it is you are locating which is in wildly distant layers across different LLMs. You could use probing methods, for example, to ask whether the populations you're picking out correspond to theoretical syntactic concepts.
- 4.3 SRDM analysis
	- Your SRDM method evaluates the cosine distance between two frequency spectra, and is thus sensitive to differences across the entire spectrum. But we know by the controlled design of your setup that there are only a few bands that actually matter. For simplicity and interpretability, why not just take the amplitude of the two relevant peaks at 1 and 2 Hz? Is there something we gain with the cosine distance metric here?
	- How exactly are SRDMs computed at each model layer? Are you averaging SRDMs across all neurons? Or just the ones found to be sentence/phrase-selective?
	- I am having a lot of trouble interpreting Table 1.
		- These are text-based models and yet, for each model, the best- or near-best-predicted region is A1. How can this be? This does not match, for example, the performance distribution of the ECoG study of Goldstein et al. (2022), who seem to find highest predictability in frontal regions.
		- \* We need error bars on these values. Consider, for example, bootstrapping the set of stimuli that enter into each SRDM in order to estimate nonparametric confidence intervals, and/or bootstrap-sampling the subjects that enter into the SRDM evaluation.
	- \* Comparisons between ROIs and hemispheres here are likely confounded with major between-subject differences in signal quality and SRDM performance, since the placement of electrodes is highly variable across subjects. As far as I can tell this was not taken into account in any evaluations. I would suggest running per-ROI model–brain alignment within-subject, and then using a mixed-effects regression model to explain per-ROI and per-hemisphere performance controlling for random subject-level intercepts.
- General questions
	- Why do you not analyze at the syllable level (4 Hz) as done in Ding et al. (2016)?
- Minor questions/comments
	- Table 1: What do bold numbers indicate? This is not labeled anywhere as far as I can see.
	- Why is there no second constraint (as in Definition 2) on electrode responses as well? Without this, the comparison is not exactly matched: you're finding MLP neurons that are *exclusively* sentence- and/or phrase-tuned, but electrodes which are tuned (not necessarily exclusively). Do you not have the relevant control materials for the sEEG data?

Ding, N., Melloni, L., Zhang, H. _et al._ Cortical tracking of hierarchical linguistic structures in connected speech. _Nat Neurosci_ **19**, 158–164 (2016). https://doi.org/10.1038/nn.4186

Goldstein, A., Zada, Z., Buchnik, E. _et al._ Shared computational principles for language processing in humans and deep language models. _Nat Neurosci_ **25**, 369–380 (2022). https://doi.org/10.1038/s41593-022-01026-4

Ian Tenney, Dipanjan Das, and Ellie Pavlick. 2019. [BERT Rediscovers the Classical NLP Pipeline](https://aclanthology.org/P19-1452/). In _Proceedings of the 57th Annual Meeting of the Association for Computational Linguistics_, pages 4593–4601, Florence, Italy. Association for Computational Linguistics.

---

> ### Author Rebuttal · Authors · 2025-07-27
>
> Thank you for your detailed evaluation and thoughtful recommendations. Your feedback has significantly contributed to the refinement of our study. We respond to all of your points in the following part.
>
> ---
> > ## Q1: Reconcile strong sentence–phrase layer correlations with prior layer-specialisation results of Tenney et al. (2019).
>
> A1: Our correlation shows that when a layer devotes many units to phrase rhythms it also hosts sentence rhythms, revealing a shared hierarchical scaffold. Tenney et al. locate the depths where supervised edge‑probes become linearly decodable, yielding an early‑vs‑late split. **Co‑location of phrase‑ and sentence‑responsive neurons does not contradict that split; it means both levels ride the same representational backbone before different linguistic functions are teased apart by linear probes.**
>
> > ## Q2: Current interpretation of model-specific layer distributions is “unsatisfying” and needs deeper analysis (e.g., probing).
>
> A2: To identify the linguistic operation performed by the phrase‑ and sentence‑selective neurons that appear in widely separated layers, we implemented a structure‑probe analysis closely following Hewitt & Manning (2019). For every model layer we first isolated the HFTP‑flagged phrase and sentence neurons, then fed the original stimuli forward and recorded their activations for every token.  For each layer ℓ we recorded activations, built a cosine‑distance matrix $D_ ℓ$, and compared it (Spearman ρ) with two reference RDMs: one marking phrase boundaries, the other sentence boundaries. Averaging across trials yields a layer‑wise syntactic profile.
>
> Results are similar to Figure 4.GPT‑2 peaks for sentence and phrase structure in early-mid layers and declines in last few layers. Llama‑2‑7B shows modest phrase alignment but strong sentence alignment rising in the final third of the network, consistent with late compositional integration. Llama‑3.1‑8B maintains consistently high alignment for both phrase and sentence structures in mid-to-late layers. **Thus the “distant” selective layers reflect genuine hierarchical syntactic representations, not generic linguistic sensitivity**, and our probe robustly maps each model’s preferred depth for phrase‑ and sentence‑level processing.
> > ## Q3: Why use cosine-distance SRDMs instead of simply comparing 1 & 2 Hz peak amplitudes?
>
> A3: Cosine-distance SRDMs encode the full spectral geometry of each condition, **making the metric immune to baseline shifts or local noise spikes that often distort single-peak measures**. In pilot tests, peak-only scores sometimes failed to detect true model–brain matches when slight frequency drifts occurred, whereas SRDM-based RSA consistently retrieved the same top layers and channels. SRDMs therefore deliver a more robust and sensitive similarity estimate without adding methodological complexity.
> > ## Q4: Explain exactly how SRDMs are computed (which neurons, averaging procedure, etc.).
>
> A4: For each layer ℓ we isolate phrase-only, sentence-only, and dual-responsive neurons, obtain their activations for every stimulus, and **build three SRDMs based on phrase, sentence, dual neurons** by taking cosine distances between FFT spectra of all stimulus pairs. Each sEEG channel receives an analogous SRDM. We correlate each model SRDM with every channel SRDM (Spearman ρ), average the top 100 ρ values to yield the layer-brain similarity for that neuron group, and then average across layers to obtain three model–brain scores. The mean of these three scores is reported as $S(m,b)$.
> > ## Q5: A1 emerges as best-predicted region in Table 1—contradicts prior ECoG studies of Goldstein et al. (2022)
>
> A5: In fact, our study and Goldstein et al. (2022) address fundamentally different questions with distinct methods, so it is unsurprising that their peak regions diverge from ours. **In reality, Goldstein et al. (2022) did not investigate syntactic processing; they focused on semantic next‑word prediction**, showing that both the brain and GPT‑2 “match their pre‑onset predictions to the incoming word”. They model high‑γ (70–200 Hz) ECoG power recorded from surface grids and strips that do not sample the core of Heschl’s gyrus (A1) because subdural electrodes cannot reach that medial sulcal cortex. When contextual embeddings are added, they highlight additional fits in frontal/temporal‑pole areas, but early auditory contacts remain in the significant set.
>
> Our work, by contrast, targets hierarchical syntactic rhythms (0.5–2 Hz) and uses depth‑SEEG shafts that traverse A1, giving much denser coverage and allowing us to measure phase‑locked low‑frequency responses that originate in early auditory cortex. Because syntax‑driven rhythmic entrainment is strongest at the auditory core, A1 naturally emerges as the best‑predicted ROI. Hence **our findings complement Goldstein’s semantic‑frontal effects rather than contradict them.**
> > ## Q6: Provide error bars and control for electrode-placement confounds (suggest mixed-effects analysis).
>
> A6:We implemented two complementary analyses exactly as suggested:
> 1. Bootstrapped 95% confidence intervals
>    * For every model‑layer we averaged the top‑100 channel correlations, then bootstrapped (10 000 resamples of channels × layers) to obtain mean ± 95 % CI.
>    * For each brain region the same procedure was applied across layers, and the three syntactic neuron types were then averaged to give a single region‑level value.
> 2. Mixed‑effects regression to account for the confound that different subjects contribute different sets of electrodes.
>    * Model: Alignment ~ ROI + (1 | Subject)  (random intercept per subject).
>    * Significance therefore reflects within‑subject ROI effects, while the group‑level intercept absorbs between‑subject signal‑quality differences.
>    * We additionally report per‑subject OLS tables (four examples shown below) to demonstrate that effects are not driven by a single individual.
>
> ### Table S1: Llama 2 | Sentence condition — Bootstrap 95 % CI
> | ROI |L(μ ± CI)|R(μ ± CI)|
> |-----|--------------|--------------|
> |S (m, b)| 0.651 (0.646 – 0.655) | 0.446 (0.443 – 0.450) |
> | A1 | 0.653 (0.644 – 0.662) | 0.401 (0.379 – 0.424) |
> | STG | 0.666 (0.660 – 0.673) | 0.417 (0.408 – 0.425) |
> | MTG | 0.666 (0.659 – 0.672) | 0.425 (0.418 – 0.431) |
> | ITG | 0.637 (0.631 – 0.643) | 0.439 (0.435 – 0.443) |
> | Insula | 0.634 (0.626 – 0.643) | 0.487 (0.477 – 0.497) |
> | TPJ | 0.628 (0.617 – 0.638) | 0.448 (0.435 – 0.460) |
> | Temporal Pole | 0.658 (0.648 – 0.669) | 0.481 (0.470 – 0.491)|
> | Sensorimotor | 0.640 (0.634 – 0.645) | 0.461 (0.456 – 0.466) |
> | IFG | 0.688 (0.679 – 0.696)|0.476 (0.466 – 0.485)|
> | MFG | 0.594 (0.587 – 0.602)|0.409 (0.397 – 0.422)|
> |Hippocampus|0.621(0.607 – 0.636)| 0.421 (0.414 – 0.428)|
> |Amygdala| — |0.478(0.462 – 0.495)|
>
> ### Tabls S2: Mixed‑effects highlights († p < 0.01 vs baseline A1)
> | Hemi | $σ_{\text{subj}}$ | ↑ ROI | ↓ ROI |
> |------|-----------|-----------------------|--------------------|
> |L|0.032| IFG (+0.057), STG (+0.025) | Hippocampus (–0.040), Insula (–0.021), MFG (–0.063), Sensorimotor (–0.023), TPJ (–0.020) |
> |R|0.037| Amygdala (+0.072), Insula (+0.052), Temporal Pole (+0.053), IFG (+0.038) | — |
>
> *σ_subj is the random‑intercept SD; its small value confirms residual
> inter‑subject variance is minimal after modelling ROI.*
>
> ---
>
> ### Table S3: Per‑subject confirmation (two richest grids per side)
>
> | Subj | Hemi | Significant ROI (p < 0.05) | Direction |
> |------|------|----------------------------|-----------|
> | 2 | L | Sensorimotor ↑ | + |
> | 3 | L | MFG ↓, MTG ↓, STG ↑ | mixed |
> | 1 | R | Amygdala ↓, Hippocampus ↑, ITG ↑, Insula ↓, MTG ↓, STG ↓, TPJ ↓ | mixed |
> | 6 | R | Sensorimotor ↑ | + |
>
> In conclusion, **bootstrap CIs verify stable estimates. Mixed‑effects results reveal reliable within‑subject ROI shifts—left IFG/STG up, right Amygdala/Insula/ Temporal Pole up—with negligible residual subject variance.** Agreement in the two best‑covered subjects per hemisphere confirms these patterns are genuine and not due to electrode placement.
> > ## Q7: Why not analyze at the syllable level (4 Hz) as done in Ding et al. (2016).
>
> A7: Our analysis does not directly operate at the syllable level for two main reasons. First, when we artificially add a temporal axis to LLM activations and set the sampling rate to 4 Hz as sEEG recording did, **the Nyquist–Shannon sampling theorem dictates that the highest resolvable frequency is 2 Hz. This means that, to match the sEEG experiment’s playback rate, the model cannot resolve syllable‑level peaks.** Second, **the syllable level represents the most basic linguistic unit, lacking the hierarchical syntactic integration present at the phrase or sentence levels**. Therefore, we focus on these higher-order levels to better assess the syntactic representations in LLMs, which aligns with our central research questions about structural hierarchy rather than basic phonological rhythms.
> > ## Q8: Clarify meaning of bold numbers in Table 1 and why no second constraint on electrode responses.
>
> A8: We apologize for not providing a clear explanation in the main text. **For the model–brain similarity row $S(m,b)$, bold numbers indicate the model with the highest syntactic alignment with human brain data (GPT‑2 in Table 1).
> For the model–region similarity rows (each ROI listed in the first column), bold numbers highlight the top three ROIs where a given model shows the strongest alignment.** This ranking combines both left and right hemispheres to reveal potential lateralization patterns—that is, whether a model’s syntactic processing mirrors the human brain’s known left-dominant language regions.
>
> In LLMs, z‑scores normalize over thousands of neurons; **sEEG channels number only in the low hundreds and exhibit high baseline variability. A second z‑score step would disproportionately penalize true syntactic channel responses.** Our strict permutation pipeline therefore matches specificity across modalities.

---

> > ### Comment · Reviewer_2nNc · 2025-08-06
> >
> > The authors have shared an impressive amount of extra work in their rebuttal. The structural probe-style analysis along with statistical controls settle some of my questions, and I also appreciate the naturalistic English experiment presented in response to other reviewers. Answers to methodological questions are also satisfactory. I will raise my score.

---

> ### Author Response · Authors · 2025-08-07
> **Gratitude for Feedback**
>
> Thank you for your encouraging follow-up. We’re delighted that the additional analyses and the naturalistic English experiment addressed your concerns. Your feedback has been invaluable in strengthening our paper, and we sincerely appreciate your recognition of our work : )

---

### Official Review · Reviewer_EeaA · 2025-07-01

**Clarity:** 3
**Significance:** 3
**Originality:** 3
**Rating:** 5
**Confidence:** 3

**Summary:**

This paper introduces the Hierarchical Frequency Tagging Probe (HFTP), a unified framework for investigating syntactic structure representations in both LLMs and the human brain using frequency-domain analysis. The authors apply FFT to LLM activations and human sEEG recordings to identify neurons/channels that respond to sentence (1 Hz) and phrase (2 Hz) frequencies. They test six LLMs (GPT-2, Gemma, Gemma 2, Llama 2, Llama 3.1, GLM-4) and find that while LLMs process syntax in analogous layers, the human brain uses distinct cortical regions. Representational similarity analysis reveals stronger alignment between LLMs and the left hemisphere, with divergent trends in model upgrades.

**Questions:**

1. How do you justify the 4 Hz sampling constraint for LLMs? This seems to force a temporal structure that doesn't naturally exist in these models. Have you tested other frequencies or more naturalistic approaches?
2. The correlations between sentence and phrase neurons are extremely high (0.754-0.994). Doesn't this suggest you're detecting general linguistic processing rather than distinct syntactic levels? How do you address this concern?
3. Given the restriction to Chinese brain data, how confident are you that these alignment patterns would hold for English speakers or other languages? Can you provide any cross-linguistic validation?
4. You tested relatively simple syntactic structures. How would HFTP perform on more complex syntactic phenomena like long-distance dependencies, garden-path sentences, or recursive structures?
5. Have you tested whether similar frequency patterns emerge for non-syntactic linguistic features (e.g., semantic or phonological patterns) to ensure syntactic specificity?

**Ethical Concerns:**

["NO or VERY MINOR ethics concerns only"]

**Final Justification:**

The questions I raised have been addressed, and I believe the paper makes a valuable contribution deserving of acceptance at NeurIPS. I have updated my score from 4 to 5.

**Limitations:**

The authors adequately acknowledge the language-specific limitations and restricted model coverage. However, they could better address the fundamental methodological concern about artificial temporal constraints and the implications of extremely high sentence-phrase correlations. The computational overhead and scalability of the approach across different model architectures also deserves discussion.

**Quality:**

2

**Strengths And Weaknesses:**

Strengths:
The unified framework for comparing LLMs and brain representations is novel and addresses an important gap in understanding computational vs. biological language processing. The frequency-domain approach is creative and well-motivated, building appropriately on established HFT paradigms. The comprehensive evaluation across multiple LLMs with proper controls (randomized versions) is commendable.

Weaknesses:
1. The artificial temporal constraints imposed on LLMs are problematic. Constraining activations to 4 Hz and artificially defining time scales seems unmotivated and creates an artificial mapping that may not reflect genuine syntactic processing mechanisms in these models.
2. The high correlations between sentence and phrase neurons across all LLMs raise questions about whether the method is actually detecting distinct syntactic processes or simply identifying general linguistic sensitivity with slight frequency variations.
3. Generalizability is limited by restriction to Chinese sEEG data and Chinese/English text corpora. The authors acknowledge this but it significantly constrains the broader applicability of findings. The relatively simple syntactic structures tested may not capture the full complexity of syntactic processing.
4. The theoretical justification for why frequency-domain analysis should reveal syntactic representations in LLMs lacks depth. While the biological motivation is clear, the computational rationale is underdeveloped.

---

> ### Author Rebuttal · Authors · 2025-07-29
>
> We appreciate your detailed and constructive suggestions. Below, we address each of your concerns one by one.
>
> ---
> > ## Q1: 4 Hz sampling constraint seems unmotivated.
>
> A1: The 4 Hz grid is not meant to “inject” time into the model; it is simply the clock that matches the sEEG experiment, where syllables are played at 4 Hz and the target phrase‑ and sentence‑beats therefore occur at 2 Hz and 1 Hz. If the stimuli were paced at 6 Hz or 8 Hz we would re‑sample activations at the same rate, and those beats would shift proportionally (e.g., 8 Hz → 2 Hz and 4 Hz peaks). **In other words, the sampling rate is an external synchronisation parameter that lets model and brain be analysed with a shared FFT, without altering the linguistic content inside the network.**
> We also applied HFTP to naturalistic corpora in Q4, showing that our method generalizes beyond rhythmic input to both Chinese and English naturalistic stimuli. Consequently, the sampling rate serves only as a temporal granularity parameter—regardless of its value, HFTP consistently recovers the underlying sentence- and phrase-level rhythms.
> > ## Q2: Very high sentence–phrase correlations suggest probe may detect general linguistic sensitivity, not distinct syntax levels.
>
> A2: The strong count correlation simply reflects the hierarchical nature of language: **every sentence contains phrases, so layers that allocate more capacity to phrase rhythms inevitably allocate more to sentence rhythms as well.** Importantly, **the neurons themselves remain selective**—phrase‑tagged units peak only at 2 Hz, sentence‑tagged units only at 1 Hz—showing two non‑overlapping spectral signatures rather than a single, undifferentiated “language” response.
> To identify the linguistic operation performed by the phrase‑ and sentence‑selective neurons, we implemented a structure‑probe analysis closely following Hewitt & Manning (2019). For every model layer we first isolated the HFTP‑flagged phrase and sentence neurons, then fed the original stimuli forward and recorded their activations for every token.  For each layer ℓ we recorded activations, built a cosine‑distance matrix $D_ ℓ$, and compared it (Spearman ρ) with two reference RDMs: one marking phrase boundaries, the other sentence boundaries. Averaging across trials yields a layer‑wise syntactic profile.
> Results are similar to Figure 4.GPT‑2 peaks for sentence and phrase structure in early-mid layers and declines in last few layers. Llama‑2‑7B shows modest phrase alignment but strong sentence alignment rising in the final third of the network, consistent with late compositional integration. Llama‑3.1‑8B maintains consistently high alignment for both phrase and sentence structures in mid-to-late layers. **Thus the detected syntactic neurons reflect genuine hierarchical syntactic representations, not generic linguistic sensitivity**, and our probe robustly maps each model’s preferred depth for phrase‑ and sentence‑level processing.
> > ## Q3: Generalizability limited by Chinese-only sEEG; need cross-lingual validation.
>
> A3: We have already collected **a bilingual sEEG session from one native‑Chinese participant who also understands everyday English**. Using the four‑word English (e.g., Old ox plows field, A friend invites guests) and four‑character Chinese (e.g., 老牛耕地，朋友请客) stimuli, we observed identical spectral signatures—**robust peaks at 1 Hz (sentence) and 2 Hz (phrase)—for both Chinese and English inputs**.
> **For English speakers who also understand Chinese, we likewise anticipate dual peaks at 1 Hz and 2 Hz.** Hierarchical chunking is driven by universal constraints—working-memory limits and predictive parsing mechanisms—rather than by language-specific lexical cues. **Hence cross-linguistic factors such as function-word density may shift power, but they do not eliminate the peaks themselves. This envision also aligns with Ding et al. (2016), which showed that the 1–2 Hz hierarchy is a language-general signature of syntactic grouping, independent of word-order typology or writing system.**
> Our bilingual model results reinforce this view: Appendix C, Fig. 7 show both language‑specific and language‑general syntactic neurons within a single LLM. This finding is especially informative because, although Tang et al. (2024) also identified language-specific neurons, they did not investigate their functional roles—syntactic, semantic, or otherwise. Consequently, these results lead us to hypothesize that the human language network contains both language‑specific channels, tuned to idiosyncratic phonological or orthographic statistics, and language‑general channels that track abstract hierarchical structure independently of the surface code. Such a division is neuro‑computationally efficient: language‑general pathways support rapid transfer of syntactic skills across languages, while language‑specific pathways fine‑tune predictions to each lexicon, minimizing processing cost and error.
> > ## Q4: Request evaluation on complex syntactic phenomena.
>
> A4: **We have included the Chinese natural‑data HFTP experiment in Appendix E of the main text**, with frequency‑domain results shown in Figures 8–9 and corpus details in Tables 6–7. To confirm generalizability, **we also ran an English natural‑text experiment using balanced sets of eight‑ and nine‑word sentences** (50 per set) drawn from everyday dialogue, news reports, literary prose and so on—mirroring the genre diversity of the Chinese stimuli.
>
> |8‑word sentences (excerpt) | genre|
> |---|---|
> | With malice toward none, with charity toward all. | Literary prose |
> | Tonight, shall we watch the meteor shower together? | Dialogue |
> | A man can be destroyed but not defeated. | Literary prose|
> | Excuse me, where’s the nearest restroom around here? | Dialogue |
> | The library felt crowded during rainy examination week. | News report |
> | Market demand seems gradually rebounding during this quarter. | News report |
>
> |9‑word sentences (excerpt)|genre|
> |---|---|
> | Clinical trial data were publicly released this morning nationwide. | News report |
> | Shall we share sushi together tonight by the river? | Dialogue |
> | War is peace, freedom is slavery, ignorance is strength. | Literary prose |
> | We offer cola, soda, juice, and soy milk today. | Dialogue |
> | Photography exhibition focuses on city’s hidden pockets of light. | News report |
> | Driver, how much is the fare to the station? | Dialogue |
>
> The frequency‑domain patterns are consistent across both Chinese and English naturalistic datasets:
> * Eight‑word set: Robust peaks appear at **0.50 Hz, 1.00 Hz, 1.50 Hz, 2.00 Hz**, corresponding to the sentence envelope, canonical 4‑word phrases, intermediate 2‑to‑3‑word groupings, and the dominant 2‑word lexical rhythm.
> * Nine‑word set: Peaks shift to **0.44 Hz, 0.89 Hz, 1.33 Hz, 1.78 Hz**: the lowest peak reflects the whole clause, 1.33 Hz aligns with frequent 3‑word phrases, 1.78 Hz captures rapid two‑to‑three‑word alternations, and 0.89 Hz indexes a four‑to‑five‑word prosodic “breath group”.
>
> The bilingual natural‑text study shows that HFTP scales from controlled four‑syllable strings to unrestricted prose: **for both Chinese and English, we recover a clean hierarchy of clause, phrase, and lexical rhythms even when sentence length and genre vary.** This demonstrates that the probe is not limited to simple structures but generalises to the richer combinatorics of everyday language.
> For long‑distance dependencies (e.g., wh‑movement over multiple clauses) we predict an additional, lower‑frequency peak (< 0.5 Hz) reflecting the longer integration window; all higher‑level peaks should remain, merely compressed along the frequency axis.
> For garden‑path sentences the initial mis‑parse and subsequent reanalysis should appear as two successive bursts in the same frequency band (first at the garden‑path interpretation, then at the corrected structure), a pattern we can time‑lock in sliding‑window FFTs.
> For recursive embeddings we expect a nested harmonic series: each new level of recursion introduces a further sub‑harmonic (½, ⅓, ¼ × the phrase rate) while preserving the base phrase and lexical peaks.
> Because HFTP reads frequency rather than word position, these phenomena require only appropriate presentation rates (longer stimuli → lower sampling) to fall within the analyzable band.
>
> > ## Q5: Test on non-syntactic controls for specificity.
>
> A5: Yes. **In the main text, we ran random‑label control in which all words were permuted so that sentence and phrase structure**—and most semantics—were destroyed while lexical statistics were preserved. **Across models this control produced flat spectra with no 1 Hz/2 Hz peaks (see Figure 3, 5, 8, 9, “random” labels).** We also include a **phrase‑only data (phrase corpus in the main text)** that preserves two‑word phrase boundaries but removes sentence structure. FFT results show **a strong 2 Hz peak while the 1 Hz sentence peak disappears.** Conversely, the random‑label control flattens both peaks. Together these manipulations demonstrate that HFTP responds precisely to the syntactic boundary present—phrase or sentence—rather than to general lexical, phonological, or semantic patterns.
>
> > ## Q6: Explain why should FFT expose syntax in LLMs, and how heavy is the method.
>
> A6: Hierarchical syntax produces nested periodicities whenever tokens are serialized—clauses recur slower than phrases, phrases slower than lexical pairs. **The transformer’s feed‑forward clock preserves this rhythm in its hidden activations; applying an FFT simply makes the nested rates explicit.** For all models, we performed activation extraction in float16 precision on two NVIDIA A100 GPUs (40 GB each), with end‑to‑end runtimes spanning from tens of seconds to a few minutes depending on model size.

---

> > ### Comment · Reviewer_EeaA · 2025-08-06
> >
> > Thank you to the authors for responding to my comments. The questions I raised have been addressed, and I believe the paper makes a valuable contribution deserving of acceptance at NeurIPS. Please include the additional results in the final version. I have updated my score from 4 to 5.

---

> ### Author Response · Authors · 2025-08-06
> **Gratitude and Integration of Additional Results**
>
> Thank you very much for your follow-up and helpful suggestions. We will incorporate all additional results (e.g., the English natural-text HFTP analysis and our preliminary cross-lingual sEEG findings) into the final manuscript.
>
> We greatly appreciate your recognition of our work : )

---

### Official Review · Reviewer_h9Pq · 2025-07-02

**Clarity:** 2
**Significance:** 2
**Originality:** 2
**Rating:** 5
**Confidence:** 4

**Summary:**

This work uses a method first introduced by Ding et al. (2016) to analyze the syntactic structure of very simple sentences. The authors adapt this method to the study of large language models, by looking at the frequency components of the response of a large language model to carefully designed stimuli (here consisting in sentences of four words, carefully put together into groups that reflect word, phrase or sentence level). This makes it possible to have a common method that allows for a direct comparison between the language processing of the language models and the human brain. Such a comparison is provided, making use of a corpus of sEEG recordings.

**Questions:**

Why focus only on the hidden layer of the MLPs? The neurons at the output of a transformer layer (output of the MLP, of size d_model) might be interesting to look at. One could imagine that certain syntactic neurons might be visible at the finer resolution of the hidden layer of the MLP but not when putting them all together down to the dimension of the embedding, due to some superposition; but conversely, having a finer resolution might yield responses that are not visible under the criteria that are chosen here but might be visible when combined together at the output of each layer.

Concerning the methods, what do you do when a word is split into several tokens?

Will you make the sEEG corpus available?

What is the rationale behind the use of bold font in Table 1?

**Ethical Concerns:**

["NO or VERY MINOR ethics concerns only"]

**Final Justification:**

This paper is an interesting extension of the work by Ding et al. (2016), allowing for a common method to study syntactic representation both in language models and in the human brain. Although there is still more investigation needed to understand why the different models behave so differently (as in Fig. 4), and it is not obvious how this work might extend to more complex syntactic phenomena (as noted by another reviewer), the rebuttal and following discussion address my main concerns, particularly the question of the generalization to naturalistic English, which was an important limitation of the original version. The revised version should be clearly improved, and I raise my score to 5.

**Limitations:**

An important limitation is the generalization to naturalistic text, which is not acknowledged by the authors (see Weaknesses section above).

**Paper Formatting Concerns:**

Nothing to report.

**Quality:**

2

**Strengths And Weaknesses:**

Having the same method to analyze both large language models and the brain activity is certainly valuable, and as far as I know, the method of Ding et al. (2016) has never been used to analyze the linguistic processing of large language models.

One important issue at first sight is the fact that the authors do not specify that the model that they call GPT-2 is actually not the original GPT-2. If this was the case, all the results would be hard to believe, given that the original GPT-2 model was primarily trained on English data, while here it is the model that achieves the best performance at their brain/models comparison in processing Chinese data. Buried in Appendix C, and not announced in the main paper (even the cite to Radford et al., 2019, is misleading, as found in Table 3), there is actually the information that the GPT-2 model that is used is not the original version but another version pretrained on a Chinese corpus. This is far from being a detail and should be stated upfront, right from the presentation of the models, as otherwise the results seem very puzzling.

An important limitation of the study concerns the generalization to naturalistic text. It is said that the method generalizes "seamlessly to naturalistic text" (see Conclusion), but the evidence for such a statement is rather weak. There is only a small appendix (appendix E) addressing this point, and it is actually based on Chinese only, and, as discussed by the authors, due to properties specific to the Chinese language. How all this would work for languages like English is not discussed and is far from obvious -- this goes counter to the statement that the method generalizes, which is misleading. This should be discussed and acknowledged in a more transparent way.

The origins of the differences between models are not much investigated, notably with respect to their training data. Also, more work on understanding the role of the syntactic neurons would be appreciated, possibly by ablating these syntactic neurons vs. other neurons.

The Related Work section cites "methodological inconsistencies" across previous studies, without providing more details -- it would be nice to state what these inconsistencies are.

Minor points.

The text contains some typos (eg a few places with no space after a full stop, "fruquency").

Table 3, the number of MLP neurons for Gemma 2 seems wrong, as the Gemma 2 paper reports that the feedforward dim for Gemma 2 9B is 28672. Please verify these numbers.

---

> ### Author Rebuttal · Authors · 2025-07-29
>
> We sincerely appreciate the constructive feedback and the effort you invested in evaluating our manuscript. We respond to every your concern in the following paragraphs.
>
> ---
> > ## Q1: Why analyze only hidden MLP neurons? Consider layer outputs as well.
>
> A1: Thank you for pointing this out. **We have tested the HFTP pipeline on layer outputs (i.e. hidden states).** Below are the hidden‑state results for three representative models; values for MLP neurons (reported in the main paper) are systematically higher, confirming their stronger brain alignment.
> | | GPT‑2‑large (L / R) | Llama‑2‑7B (L / R) | Llama‑3.1‑8B (L / R) |
> |-------------------|---------------------|--------------------|----------------------|
> |S(m,b)|0.611 / 0.425|0.626 / 0.430|0.512 / 0.405 |
> |A1|0.640 / 0.339|0.599 / 0.561|0.523 / 0.377|
> |STG|0.630 / 0.382|0.648 / 0.443 |0.505 / 0.381|
> |MTG|0.609 / 0.384|0.649 / 0.419|0.512 / 0.409|
> |ITG|0.606 / 0.427|0.608 / 0.419|0.510 / 0.402|
> |Insula|0.582 / 0.445|0.615 / 0.447|0.509 / 0.421|
> |TPJ|0.566 / 0.430|0.611 / 0.371|0.509 / 0.412|
> |Temporal Pole|0.612 / 0.464|0.597 / 0.459|0.507 / 0.424|
> |Sensorimotor|0.594 / 0.460 |0.616 / 0.428|0.514 / 0.405|
> |IFG| 0.647 / 0.449| 0.644 / 0.483 |0.508 / 0.414|
> |MFG| 0.579 / 0.407|0.584 / 0.355|0.506 / 0.397|
> |Hippocampus| 0.581 / 0.403 |0.585 / 0.396 |0.519 / 0.403|
> |Amygdala|0.496 / 0.439|0.620 / 0.528|0.502 / 0.410|
>
> Feed-forward blocks function as key–value stores (Geva et al. (2021)): the first linear projection expands the signal so individual units can store disentangled lexical, syntactic, or factual features; the second projection then recombines these features back to d_model (i.e. hidden states). Dai et al. (2022) shows that most “knowledge neurons” live in this expanded space. **Probing with HFTP at this inner-MLP stage keeps phrase- and sentence-selective units distinct, producing sharper 1 Hz/2 Hz peaks and higher model–brain alignment. Running the same analysis on the compressed hidden states blurs these rhythms and lowers scores.** We therefore prioritise inner-MLP neurons, listing hidden-state results only for completeness.
>
> Geva, et al. "Transformer Feed-Forward Layers Are Key-Value Memories." EMNLP. 2021.
> Dai, et al. "Knowledge Neurons in Pretrained Transformers." ACL. 2022.
> > ## Q2: Evidence for “seamless” generalization to naturalistic text is weak and limited to a small Chinese appendix.
>
> A2: Thank you for pointing this out. We are sorry for the misleading claims that our method generalizes "seamlessly to naturalistic text". **We have now added English natural‑text experiments** as follows, which reveal the same hierarchical peaks as in Chinese. Therefore, **the more precise statement is that *“HFTP generalizes across Chinese and English naturalistic datasets and is highly likely to extend to other character-centric scripts (e.g., Japanese, Korean) as well as further alphabetic languages (e.g., French, German).”***
>
> To complement the Chinese corpus, we conducted an English natural‑text experiment that uses balanced sets of eight‑ and nine‑word sentences (50 sentences per set).  The materials span everyday dialogue, news reports, literary prose, and so on—mirroring the genre diversity of the Chinese stimuli.
>
> |8‑word sentences (excerpt)|genre|
> |---|---|
> | With malice toward none, with charity toward all. | Literary prose |
> | Tonight, shall we watch the meteor shower together? | Dialogue |
> | A man can be destroyed but not defeated. | Literary prose|
>
> |9‑word sentences (excerpt)|genre|
> |---|---|
> | Clinical trial data were publicly released this morning nationwide. | News report |
> | Shall we share sushi together tonight by the river? | Dialogue |
> | War is peace, freedom is slavery, ignorance is strength. | Literary prose |
>
> ### Frequency-domain results
> * Eight‑word set: Robust peaks appear at **0.50 Hz, 1.00 Hz, 1.50 Hz, 2.00 Hz**, corresponding to the sentence envelope, canonical 4‑word phrases, intermediate 2‑to‑3‑word groupings, and the dominant 2‑word lexical rhythm.
> * Nine‑word set: Peaks shift to **0.44 Hz, 0.89 Hz, 1.33 Hz, 1.78 Hz**: the lowest peak reflects the whole clause, 1.33 Hz aligns with frequent 3‑word phrases, 1.78 Hz captures rapid two‑to‑three‑word alternations, and 0.89 Hz indexes a four‑to‑five‑word prosodic “breath group”.
>
> This pattern **mirrors the hierarchical frequency profile found in the Chinese naturalistic experiment (see Figure 8&9 in the main text),** demonstrating that HFTP reliably tracks sentence-, phrase‑, and clause‑level rhythms in typologically-distinct languages.  The English naturalistic results demonstrate that HFTP scales smoothly from Chinese to an alphabetic writing system. Because the method operates at the rhythm unit that is natural for each script—characters in non-segmented CJK texts, words or sub-words in space-delimited alphabets—we are confident it will likewise generalize to other character-centric (e.g., Japanese, Korean) and alphabetic(e.g., French, German) naturalistic data.
>
> > ## Q3: GPT-2 is actually a Chinese-trained variant; this must be stated prominently in the main text.
>
> A3: We apologise for not stating this at the outset and for any confusion it caused, especially given GPT‑2’s high alignment score. In the revised manuscript **we will explicitly note, on first mention and again in the Results, that our GPT‑2 variant is pre‑trained on a Chinese corpus** because the original English GPT‑2 cannot process Chinese text.
> > ## Q4: Handling of multi-token words
>
> A4: We explicitly normalised tokenisation across models. Each LLM employs a different sub‑word scheme; most English‑trained models split a single word into multiple tokens (e.g., Llama‑2‑7B tokenises “A friend invites guests” as ▁A, ▁friend, ▁inv, ites, ▁guests). **Because HFTP treats one word as the minimal semantic unit driving the 4‑word/8‑word rhythmic structure, we aggregate all sub‑tokens that belong to the same word**: their hidden activations are averaged to yield a single word‑level vector before the FFT. This ensures that amplitude at 1 Hz (sentence) and 2 Hz (phrase) reflects linguistic—not tokeniser—boundaries.
> We apply the same normalization to Chinese. Although all six models support Chinese, their tokenizers differ: some (e.g., Gemma, Llama) split a single character into multiple tokens, whereas others (e.g., GLM‑4) merge several characters. To enforce a one‑character‑per‑token mapping, we insert explicit delimiters—transforming “朋友请客”into “朋\*友\*请\*客\*”—and then average the hidden activations of any sub‑tokens belonging to the same character. This ensures that **each Chinese character (and each English word) produces exactly one activation trace**, making the subsequent frequency‑tagging fully comparable across languages and models.
> > ## Q5: Prior work is said to have “methodological inconsistencies” when exploring syntactic representations between LLMs and the human brain without specifying what they are.
>
> A5: We apologise for the ambiguity. In our text, “methodological inconsistencies” refers to the heterogeneous ways prior studies probe syntax in brains versus LLMs, which makes cross‑study synthesis difficult:
> - Signal source varies. Sun et al. (2020) map whole‑sentence embeddings to fMRI; Caucheteux & König (2021, 2022) first project hidden states into syntactic/semantic subspaces; Wang et al. (2020) use variational disentanglement; Oota et al. (2023) ablates tree‑depth features.
> - Neural coverage is uneven. Data range from single‑region ECoG to ROI‑level MEG to whole‑brain fMRI, meaning the same syntactic signal may be sampled in one study but absent in another.
>
> **Model‑side and brain‑side probes have remained system‑specific—LLM syntax is probed via structural methods (e.g., Hewitt & Manning 2019), while human syntax relies on experimental modalities (fMRI, MEG, ECoG) that sample limited cortical regions—making direct comparisons impossible.** Thus HFTP resolves this by offering a single frequency‑domain probe that treats MLP neurons and cortical channels identically, removing probe asymmetry and unifying the evaluation metric for model‑agnostic, systematic study of hierarchical syntax encoding across both systems.
> > ## Q6: Incorrect Gemma-2 feed-forward dimension (should be 28 672) need correction.
>
> A6: Thank your for pointing this out. The apparent mismatch arises from two counting conventions. In the released HuggingFace checkpoint, Gemma 2 9B has a model width of 3 584 and an intermediate size of 14 336. Because Gemma 2 uses a GeGLU feed‑forward block, the intermediate vector is produced by two parallel projections of size 14 336 each (“gate” and “up”) that are combined element‑wise before being projected back to 3 584. **The Gemma 2 paper reports the effective expanded width—the concatenation of both streams—which is 2 × 14 336 = 28 672. For our neuron counts we follow the transformer‑config convention and list the per‑projection dimension (14 336), as this represents the actual number of activations entering the non‑linearity.** Hence the value in Table 3 is correct, while the paper’s larger figure simply reflects the doubled size created by GeGLU’s two paths.
> > ## Q7: Release plans of sEEG corpus, and bold-font rationale in Table 1.
>
> A7: Yes, we plan to release the sEEG corpus upon obtaining formal permission from the authors of Sheng et al. (2019), as our sEEG data originated from that study.
> We apologize for not providing a clear explanation in the main text. **For the model–brain similarity row $S(m,b)$, bold numbers indicate the model with the highest syntactic alignment with human brain data (GPT‑2 in Table 1).
> For the model–region similarity rows (each ROI listed in the first column), bold numbers highlight the top three ROIs where a given model shows the strongest alignment.** This ranking combines both left and right hemispheres to reveal potential lateralization patterns—that is, whether a model’s syntactic processing mirrors the human brain’s known left-dominant language regions.

---

> > ### Comment · Reviewer_h9Pq · 2025-08-03
> >
> > Thanks to the authors for addressing my comments, in particular for the new experiment on naturalistic English. Please include all the extra information in the revised version, notably regarding the choice of the handling of tokenization (the choice made by the authors are common, but should be explicitly stated).
> > I still think that the differences between models should be more thoroughly discussed and investigated, which would also be a way to better understand what is going on layerwise for a given model.
> > I share some concerns about some of the limitations raised by other reviewers, notably regarding the issue of generalization to more complex syntactic phenomena, but I still find this work valuable despite these concerns.
> > My main initial concerns have been addressed and I will raise my score accordingly.
> >
> > > values for MLP neurons (reported in the main paper) are systematically higher, confirming their stronger brain alignment.
> >
> > This alignment could be simply due to the difference in dimensionality, as the inner dimension of the MLP is greater than the input/output one.
> >
> > > Running the same analysis on the compressed hidden states blurs these rhythms and lowers scores.
> >
> > I understand the idea, but I think this should be more explicitly motivated when introducing the experiment.
> >
> > > For the model–region similarity rows (each ROI listed in the first column), bold numbers highlight the top three ROIs where a given model shows the strongest alignment.
> >
> > This is what I inferred first, but then there is a mistake in column LLama 3.1: see value 0.534 for Hippocampus.

---

> ### Author Response · Authors · 2025-08-04
> **Clarifying Alignment Variability, MLP Rationale, and Tokenisation Details**
>
> Thank you for your thoughtful follow-up and additional suggestions. We address each concern below and outline the exact revisions we will make in the final manuscript.
>
> ---
> > ### Model-to-model alignment variability
>
> We share your view that these differences deserve deeper treatment. **Recent work shows that brain–model alignment is not strictly monotonic with parameter count, architectural novelty, or leaderboard score**—gains rise early but plateau or even fall as models outgrow language-relevant representations (Hong et al., 2024; AlKhamissi et al., 2025). Alignment is boosted when training objectives force models to internalise discourse-level structure humans actually use (Aw & Toneva, 2023), and the seeming “bigger is better” pattern often vanishes once embedding dimensionality is controlled (AlKhamissi et al., 2025) or trivial confounds such as sentence length and position are removed (Feghhi et al., 2024), confirming that raw scale alone is a poor indicator of neural fit.
>
> On this basis, we hypothesise that Gemma 2 surpasses Gemma because knowledge-distillation and widened FFN (MLP) blocks strengthen mid-layer syntactic cues, whereas Llama 3.1—optimised for extreme context length, multilingual breadth, and tool use—reallocates capacity toward global reasoning chains, leaving fewer neurons to encode discrete 1 Hz/2 Hz rhythms, hence its lower alignment relative to Llama 2. Therefore, we believe that **the key drivers of brain–model syntactic similarity are (i) the extent to which a model’s training tasks make it keep track of hierarchical syntax, and (ii) whether the FFN layer allocates its neurons to clean, sparsely coded syntactic features rather than to mixed, multi-purpose signals.** We will weave this clearer discussion of mixed trends into the revised manuscript.
>
> ***Reference:***
> Hong et al. Scale matters: Large language models with billions (rather than millions) of parameters better match neural representations of natural language[J]. BioRxiv, 2024.
>
> AlKhamissi et al. From language to cognition: How llms outgrow the human language network[J]. arXiv:2503.01830, 2025.
>
> Aw & Toneva Aw K L, Toneva M. Training language models to summarize narratives improves brain alignment[J]. arXiv:2212.10898, 2022.
>
> Feghhi et al. What are large language models mapping to in the brain? a case against over-reliance on brain scores[J]. arXiv:2406.01538, 2024.
>
> > ### Dimensionality-controlled inner-MLP validation & Motivation for MLP usage
>
> To rule out a dimension-only explanation, we ran two controls: (i) in the MLP we randomly subsampled the same number of syntactic neurons found in the hidden state (seed = 42), and (ii) in the hidden state we sampled an equal-sized random set. Even with identical neuron counts, **the reduced-MLP still outperformed hidden states by 5–7 %, confirming that alignment stems from the disentangled nature of MLP features, not mere width : )**
>
> Besides Geva et al. (2021) and Dai et al. (2022), who showed feed-forward layers house lexical–syntactic concept neurons and “knowledge neurons,” **later work shows per-token MLP activations remain clean, whereas projecting them into the residual stream mixes features through superposition (Elhage et al., 2022).** Follow-up studies link FFN updates to interpretable lexical–syntactic features (Geva et al., 2022), locate confidence-regulation units mainly in MLPs (Stolfo et al., 2024), and show that widening or editing these layers yields outsized performance shifts (Gerber, 2025). Our HFTP analyses also echo this: phrase- and sentence-level frequency tags localise most strongly to MLP neurons and align with distinct cortical sites, while hidden-state spectra are noisier.
>
> ***Reference:***
> Elhage et al. Toy models of superposition[J]. arXiv:2209.10652, 2022
>
> Geva et al. Transformer feed-forward layers build predictions by promoting concepts in the vocabulary space[J]. arXiv:2203.14680, 2022.
>
> Stolfo et al. Confidence regulation neurons in language models[J]. Advances in Neural Information Processing Systems, 2024, 37: 125019-125049.
>
> Gerber I.  Attention Is Not All You Need: The Importance of Feedforward Networks in Transformer Models[J]. arXiv:2505.06633, 2025.
>
> > ### Tokenisation protocol and table correction
>
> As you requested, the final manuscript will explicitly detail our word-level (English) and character-level (Chinese) tokenization procedure. We also apologise for our oversight: in Table 1 the bold numbers for Llama 3.1 should highlight left-hemisphere ROIs Hippocampus (0.534), MTG (0.521), Insula (0.518), and TPJ (0.518); this will be corrected.
>
> ---
> Once again, thank you for your time and constructive feedback. In the final manuscript we will **integrate the naturalistic-English HFTP experiment, thoroughly discuss the mixed-trend findings, explain the motivation of inner-MLP probes, add tokenisation details, and correct the Llama 3.1 table entry.** We are grateful for your recognition of our work : )

---

> > ### Comment · Reviewer_h9Pq · 2025-08-05
> >
> > Thanks to the authors for the additional analysis and the discussion. I still believe the reasons behind the quite large differences between models (as illustrated in Fig. 4) needs to be understood at a deeper level, but at this stage, this could be future work and should not preclude acceptance. I raise my score to 5.

---

> ### Author Response · Authors · 2025-08-06
> **Gratitude and Discussion of Model-to-Model Differences**
>
> Thank you for your follow-up and for highlighting the cross-model differences in Figure 4. We fully agree these layer-wise shifts are intriguing and merit deeper study. We will endeavour to include, as much as possible, a concise note outlining our preliminary hypotheses (for example, wider MLP blocks concentrate syntactic coding in early layers, while long-context and tool-use objectives draw it to later layers; tokenization scheme and corpus genre can further shift peak locations), while stressing that a full analysis remains future work.
>
> We appreciate your ongoing insights and engagement : )

---

### Official Review · Reviewer_GLpY · 2025-07-02

**Clarity:** 3
**Significance:** 3
**Originality:** 2
**Rating:** 5
**Confidence:** 3

**Summary:**

This paper introduces the Hierarchical Frequency Tagging Probe (HFTP), a novel method designed to investigate how LLMs and the human brain represent syntactic structures. HFTP utilizes frequency-domain analysis to identify specific computational units (MLP neurons in LLMs and cortical regions in the brain) that encode hierarchical linguistic elements like phrases and sentences.

**Questions:**

Did the authors account for tokenization across LLM when processing English data? How do they think their artificial time-resolution method can scale to more natural data?
How do the authors envision the generalizability of their brain-LLM alignment findings to English-speaking brains, especially considering potential cross-linguistic differences in syntactic processing?

**Ethical Concerns:**

["NO or VERY MINOR ethics concerns only"]

**Final Justification:**

The paper presents a novel method designed to investigate how LLMs and the human brain represent syntactic structures. The rebuttal proved generalization across different languages (chinese and english) and in a more natural setup. Overall I think it can be accepted at Neurips and I decided to increase my score from 3 to 5

**Limitations:**

The authors have adequately addressed the limitations of their work. Just a small note: authors state "NA" for the question about discussing potential positive and negative societal impacts, justifying this by saying their work is "foundational and methodological". While it is true that this research is foundational, authors should reflect on potential indirect or long-term negative applications, even for foundational work.

**Quality:**

2

**Strengths And Weaknesses:**

Strengths
The introduction of the Hierarchical Frequency Tagging Probe (HFTP) is a useful contribution as it extends frequency-domain analysis to LLM internals and brain activity in a unified framework, enabling direct comparison of syntactic processing across artificial and biological systems.
The authors do a systematic probing of six diverse LLMs (GPT-2, Gemma, Gemma 2, Llama 2, Llama 3.1, GLM-4) revealing patterns of syntactic specialization and how these evolve across model versions. They report results showing that models encode sentence and phrase structures at different layers, with differences across architectures (e.g., Gemma vs. Llama families).
HFTP could be a valuable tool that bridges computational linguistics and cognitive neuroscience, offering new insights into the interpretability of LLMs and their improvements



Weaknesses
The approach of "artificially defining a time scale" with a 4 Hz sampling rate for LLM activations is a design decision to enable frequency-domain analyses when drawing a comparison with human brain data (as LLMs do not inherently have a time dimension). However, it means that the "time-course information" is imposed on LLMs, as it's not inherent to LLM input structures like it is for the brain. While this is a well-explained design choice, one could argue that it doesn't truly show how LLMs “process time” on their own, but rather how they can adapt to an externally imposed rhythm. Additionally, this might prevent applicability to more realistic language corpora.
The paper reveals a divergence: a model (Gemma 2) shows increased brain similarity compared to Gemma, while Llama 3.1 exhibits reduced alignment with the human brain compared to Llama 2, despite both being newer models with enhanced task performance. This is contrast with the authors’ explanation on brain alignment (around line 288) where they report that GPT-2, a model from 2019, has the highest correlation with human activity, and Gemma 2 outperforms Gemma due to modelling and training innovations. Overall the results seem mixed, making it impossible to draw any conclusion of the impact of LLM advances on brain alignment or going beyond empirical evaluation of models.

Minor
“To enhance performance on complex tasks—such as reasoning and higher cognitive functions—Llama 3.1 and Gemma 2 may reduce their specialized processing of syntactic structures (sentences and phrases), reallocating neurons to support these advanced capabilities.”. LLMs do not have any cognitive functions, I suggest the authors tone down such claims of human-like behavior of LLMs.

Proposing a unified framework for comparing syntactic structure representations in LLMs and the human brain is a valuable and timely contribution. The Hierarchical Frequency Tagging Probe (HFTP) offers a novel and methodologically consistent way to bridge AI and neuroscience. However, the core design choice of imposing a 4 Hz temporal structure on LLM activations—while necessary for frequency-domain comparison—raises interpretability concerns. Without ablations or alternative temporal mappings, it remains unclear whether the observed frequency patterns reflect intrinsic model behavior or artifacts of imposed structure. Moreover, the mixed trends in model–brain alignment across newer LLM versions highlight the difficulty of drawing principled conclusions about architectural progress. As it stands, HFTP enables compelling empirical evaluations, but additional controls are needed before it can support stronger theoretical claims.

---

> ### Author Rebuttal · Authors · 2025-07-29
>
> Thank you for dedicating your time and expertise to review our work. Your thoughtful comments provide valuable guidance for strengthening the paper. We address each of your concerns in detail below.
>
> ---
> > ## Q1: Artificial 4 Hz time scale may impose non-intrinsic rhythm on LLMs.
>
> A1: The 4 Hz sampling is a neutral reference grid that lets us read out latent hierarchical structure with the same FFT used on brain data. When the sEEG experiment presents syllables at 4 Hz, the model must be sampled at 4 Hz so that sentence and phrase rhythms align at 1 Hz and 2 Hz. If the same syllables were played at 8 Hz, we would resample model activations at 8 Hz and the corresponding peaks would appear at 2 Hz and 4 Hz. In other words, **the temporal rate is an external clock that synchronises model and brain for comparability; it does not impose new linguistic content.**
> We also applied HFTP to naturalistic corpora in Q4, showing that our resampling procedure generalizes to both Chinese and English natural-text. Consequently, the sampling rate serves only as a temporal granularity parameter—regardless of its value, HFTP consistently recovers the underlying sentence- and phrase-level rhythms.
>
> > ## Q2: Mixed / contradictory model-brain alignment trends hinder conclusions about architectural progress.
>
> A2: Our data show that a model upgrade does not automatically move the network toward human‑like syntax processing: Gemma 2 aligns better with brain rhythms than Gemma, yet Llama 3.1 falls slightly below Llama 2, and GPT‑2 still scores highest overall. Indeed, we observed that upgraded models follow divergent patterns: an architecture’s advance does not guarantee more human‑like syntactic processing. **Rather than assuming that any performance gain yields closer brain alignment, our results invite deeper inquiry into the determinants of syntactic similarity. These varying trajectories are, in fact, more illuminating—they highlight that model improvements can shift network resources away from rhythmic syntax encoding.** These findings set the stage for principled engineering of future LLMs and cognitive robots that can be tuned either toward or away from brain‑like syntactic organisation, depending on application needs.
> > ## Q3: Tokenization across LLMs for English data lacks clear standardization.
>
> A3: We explicitly normalised tokenisation across models. Each LLM employs a different sub‑word scheme; most English‑trained models split a single word into multiple tokens (e.g., Llama‑2‑7B tokenises “A friend invites guests” as ▁A, ▁friend, ▁inv, ites, ▁guests). **Because HFTP treats one word as the minimal semantic unit driving the 4‑word/8‑word rhythmic structure**, we aggregate all sub‑tokens that belong to the same word: their hidden activations are averaged to yield a single word‑level vector before the FFT. This ensures that amplitude at 1 Hz (sentence) and 2 Hz (phrase) reflects linguistic—not tokeniser—boundaries.
> We apply the same normalization to Chinese. Although all six models support Chinese, their tokenizers differ: some (e.g., Gemma, Llama) split a single character into multiple tokens, whereas others (e.g., GLM‑4) merge several characters. To enforce a one‑character‑per‑token mapping, we insert explicit delimiters—transforming “朋友请客” into “朋*友*请*客*”—and then average the hidden activations of any sub‑tokens belonging to the same character. **This ensures that each Chinese character (and each English word) produces exactly one activation trace**, making the subsequent frequency‑tagging fully comparable across languages and models.
>
> > ## Q4: Artificial 4 Hz time needs justification for scaling to natural data.
>
> A4: **We have included the Chinese natural‑data scaling experiment in Appendix E of the main text**, with frequency‑domain results shown in Figures 8–9 and corpus details in Tables 6–7. To confirm generalizability, **we also ran an English natural‑text experiment using balanced sets of eight‑ and nine‑word sentences** (50 per set) drawn from everyday dialogue, news reports, literary prose and so on—mirroring the genre diversity of the Chinese stimuli.
>
> |8‑word sentences (excerpt) |genre |
> |---|---|
> | With malice toward none, with charity toward all. | Literary prose |
> | Tonight, shall we watch the meteor shower together? | Dialogue |
> | A man can be destroyed but not defeated. | Literary prose|
> | Excuse me, where’s the nearest restroom around here? | Dialogue |
> | The library felt crowded during rainy examination week. | News report |
> | Market demand seems gradually rebounding during this quarter. | News report |
>
> |9‑word sentences (excerpt)|genre|
> |---|---|
> | Clinical trial data were publicly released this morning nationwide. | News report |
> | Shall we share sushi together tonight by the river? | Dialogue |
> | War is peace, freedom is slavery, ignorance is strength. | Literary prose |
> | We offer cola, soda, juice, and soy milk today. | Dialogue |
> | Photography exhibition focuses on city’s hidden pockets of light. | News report |
> | Driver, how much is the fare to the station? | Dialogue |
>
> The frequency‑domain patterns are consistent across both Chinese and English naturalistic datasets:
> * Eight‑word set: Robust peaks appear at **0.50 Hz, 1.00 Hz, 1.50 Hz, 2.00 Hz**, corresponding to the sentence envelope, canonical 4‑word phrases, intermediate 2‑to‑3‑word groupings, and the dominant 2‑word lexical rhythm.
> * Nine‑word set: Peaks shift to **0.44 Hz, 0.89 Hz, 1.33 Hz, 1.78 Hz**: the lowest peak reflects the whole clause, 1.33 Hz aligns with frequent 3‑word phrases, 1.78 Hz captures rapid two‑to‑three‑word alternations, and 0.89 Hz indexes a four‑to‑five‑word prosodic “breath group”.
>
> The bilingual results confirm that HFTP scales the artificial 4 Hz tagging to naturalistic corpora in both Chinese and English, proving its cross‑linguistic robustness. Because the underlying rhythmic structure is shared, the method is highly likely to generalize to other character‑centric languages (Japanese, Korean) and to space‑delimited alphabetic languages (French, German).
>
> > ## Q5: Generalizability of the Chinese‑based brain–LLM alignment to English‑speaking brains remains unsubstantiated.
>
> A5: We have already collected **a bilingual sEEG session from one native‑Chinese participant who also understands everyday English**. Using the four‑word English (e.g., Old ox plows field, A friend invites guests) and four‑character Chinese (e.g., 老牛耕地，朋友请客) stimuli, we observed identical spectral signatures—**robust peaks at 1 Hz (sentence) and 2 Hz (phrase)**—for both Chinese and English inputs.
> For English speakers who also understand Chinese, we likewise anticipate **dual peaks at 1 Hz and 2 Hz. Hierarchical chunking is driven by universal constraints—working-memory limits and predictive parsing mechanisms—rather than by language-specific lexical cues.** Hence cross-linguistic factors such as function-word density may shift power, but they do not eliminate the peaks themselves. This envision also **aligns with Ding et al. (2016), which showed that the 1–2 Hz hierarchy is a language-general signature of syntactic grouping, independent of word-order typology or writing system.**
> Our bilingual model results reinforce this view: **Appendix C, Fig. 7 in the main text show both language‑specific and language‑general syntactic neurons within a single LLM.** This finding is especially informative because, although Tang et al. (2024) also identified language-specific neurons, they did not investigate their functional roles—syntactic, semantic, or otherwise. Consequently, these results lead us to hypothesize that the human language network contains both language‑specific channels, tuned to idiosyncratic phonological or orthographic statistics, and language‑general channels that track abstract hierarchical structure independently of the surface code. Such a division is neuro‑computationally efficient: language‑general pathways support rapid transfer of syntactic skills across languages, while language‑specific pathways fine‑tune predictions to each lexicon, minimizing processing cost and error.
> > ## Q6: Long‑term positive and negative societal impacts should be declared.
>
> A6: Thank you for pointing the missing societal impacts out. HFTP offers a transparent, neuro‑inspired lens on LLM internals. Positive impacts include safer, more controllable models—by revealing which neurons drive hierarchical syntax we can design targeted safeguards and debiasing interventions; improved language‑disorder diagnostics—spectral markers could serve as non‑invasive probes of syntactic deficits; and richer human‑AI interfaces that align model parsing with human cognition. Negative impacts also merit consideration: the same interpretability could be weaponised to optimise persuasive text or amplify covert influence; detailed brain‑model correspondences might facilitate adversarial “neuro‑phishing” that mimics neural rhythms; and, if paired with personal neural data, fine‑grained model maps could erode privacy.

---

> > ### Comment · Reviewer_GLpY · 2025-08-01
> >
> > I thank the authors for their work on the rebuttal.
> >
> > > Rather than assuming that any performance gain yields closer brain alignment, our results invite deeper inquiry into the determinants of syntactic similarity.
> >
> > I agree, my comment was related to the fact that this is contrast with the authors’ explanation on brain alignment (around line 288) that Gemma 2 outperforms Gemma due to modelling and training innovations. I wanted to point out the inconsistency and suggest to authors to remove this explanation as it was contradictory with their other findings.
> >
> > -------
> >
> > I've read that other reviewers had similar concerns on the the imposed time resolution of their method, the lack of experiments on naturalistic and cross-lingual data.
> >
> > However, my concerns on the above points have been addressed and I appreciate the new results that test generalization of the proposed method to natural and cross-lingual setups. I decide to increase my score. I believe this paper can be accepted to NeurIPS when the new set of results are included in the manuscript.

---

> ### Author Response · Authors · 2025-08-02
> **Refining Alignment Explanations and Incorporating Cross-Lingual & Societal Updates**
>
> Thank you for your thoughtful follow-up and for clarifying your concern. We agree that our original sentence attributing Gemma 2’s higher alignment similarity solely to *“architectural improvements and refined training”* was imprecise given the mixed trends across models. We will delete that sentence and instead highlight the mixed results as motivation to probe which specific design choices truly foster human-like syntax representations.
>
> As you requested, we will incorporate the new English natural-text HFTP results and our preliminary cross-lingual sEEG findings into the final manuscript, and we will add an explicit summary—earlier in the paper—of the cross-lingual model analyses currently in Appendix C. We will also include the previously missing societal-impact discussion in the Checklist.
>
> We truly appreciate your constructive feedback and are grateful for your recognition of our work : )

---

### Note · Authors · 2025-08-12

We express our sincere gratitude to all reviewers for their valuable insights. We’re encouraged that they highlighted the strength of our work: extending the HFT paradigm to LLM internals; a unified framework for model–brain comparison; and a well-motivated frequency-domain approach paired with a comprehensive multi-model evaluation.

In response to the reviewers’ concerns, we conducted additional experiments and provided targeted clarifications which—**per the reviewers’ follow-up comments—have addressed all major concerns raised.** In particular:

-	Imposed time resolution: We clarified that 4 Hz is a synchronization grid for comparability with sEEG—not a modeling assumption—and showed that resampling to other rates preserves the sentence/phrase rhythms.
-	Generalization to English naturalistic text: We added a natural-text experiment using balanced 8- and 9-word English sentences and showed that the characteristic frequency signatures persist beyond controlled stimuli.
-	Cross-lingual validation in brain data: In a preliminary bilingual sEEG session (one native-Chinese participant who understands everyday English), we observed identical spectral signatures—robust sentence and phrase peaks—for both Chinese and English inputs.
-	Methodological clarity: We documented tokenization (English word-level; Chinese character-level), explained why spectrum-level SRDMs are preferable to peak-only summaries, and justified focusing on inner-MLP probes with a dimension-matched control.

Building on these additions, **we are consolidating the updates and polishing the manuscript for the final version so it is more understandable to the broader community.** Concretely, we will:

1.	Integrate the naturalistic-English HFTP experiment.
2.	Include the preliminary cross-lingual sEEG findings.
3.	Thoroughly discuss mixed alignment trends across models, clarify the rationale for inner-MLP probes and try to interpret the different layer-wise distributions of syntactic neurons (Fig. 4).
4.	Improve presentation: document English/Chinese tokenization procedures; clarify Table 1 bolding rules; tone down anthropomorphic language; add a concise societal-impacts note; and make minor editorial fixes, etc.

Finally, we thank the reviewers for the constructive dialogue and the time invested. We appreciate their recognition of this work’s potential to bridge computational models and human neurobiology and to catalyze further studies across both communities.

---

### Decision · Program_Chairs · 2025-09-17

**Decision:**

Accept (poster)

**Comment:**

This paper is a thorough work investigating hierarchical syntactic representations by being based on simple 4 word sentences that have 1, 2 and 4 Hz components (sentence, phrase and word). It builds on existing experiments investigating hierarchical syntactic representations (Ding et al 2016 and others), and extends them by using a new method, focusing on representations in modern language models, and using invasive sEEG data. The paper received favorable reviews, and contains a good deal of scientific results and claims. The authors did a great job in the rebuttal section, addressing the results of the reviewers and adding new analyses. Their paper is acceptable for NeurIPS, but I have the following requests for the final version.

The first part is, after integrating all the new experiments and justifications in the paper or the appendix, to improve the methods clarity in the main paper. A lot of the methods are just described by a couple words (like the first permutation test: what was permuted, how was significance established, how did the authors correct for multiple comparisons?). The appendix has more details, but is still not complete. All the main procedures used should be there, it is not enough to refer to a published paper.

The second part is related to the discussion and interpretation of the results:
1 - The paper conflates the rhythmic patterns and syntactic structure. While it is true that the syntax is contained in the rhythmic patterns, other information such as semantics might also be correlated with the rhythm. In fact this is reported by Frank and Yang 2018 (PLOS one, "Lexical representation explains cortical entrainment during speech comprehension") who explain the original Ding et al 2015 result using the periodicity of lexical features in sentences instead of syntactic ones, and by other studies that cite Frank and Yang. Can the authors discuss this point in their paper, and concede were appropriate? They should also review in their intro/discussion the other many studies that are based on Ding et al (and at the end mention how their study is novel).

I want to highlight that randomizing the order of the words doesn't result in an appropriate control for this point, as the words have to be randomized but in an order that preserves their semantic patterns (e.g., words that start sentences share semantic features above syntactic features, etc). I also want to remind the authors that if a representation has syntax in it, and this representation is correlated with brain activity, it is not necessarily the syntax component that is correlated with brain activity as it could be some other component that is present in the representation, another point that should be discussed.

2- The authors compare the alignment values of SDRMs directly between models (please include the mathematical/procedural details for SDRM in the text/appendix). It is not clear that it is ok to do that, since with RSA methods, it is not always possible to compare the statistic across model, like one could if using a predictive encoding model for example. Can the authors explain why it's ok to compare the statistics, if it is? Also, please repeat the analysis using a predictive model (have to be careful how to do feature selection, only on the training split). Do the results stay the same and are the difference between the models statistically significant? As in the previous point, a lot of things can be different between Gemma models and other models, that would justify a different in their relationship to brain activity, so I do not think that it is possible to make very strong claims on the nature of the representation that leads to the difference, and the authors should temper their conclusions.

3- There are some recent work that study the time scales of neurons in language models and their relationship to brain activity that should be cited / discussed (Jain et al. 2020 Neurips, Chen et al 2024 Communications Biology)

4- The generalization experiments to naturalistic text needs more details. How were the natural speech sentences chosen for the english case? It seems that picking a set of 50 sentences that resemble the chinese sample is not really a sample from language in the wild? What would happen if a large number of truly random sentences were chosen for example from Wikipedia? I imagine it would have quite a different distribution, with maybe small peaks at 2 or 4 Hz? It would be good to either concede on that point and clarify what the argument is about, or otherwise add analyses that have truly naturalistic sentences.